# Emergence of transformation-tolerant representations of visual objects in rat lateral extrastriate cortex

Sina Tafazoli[1†‡], Houman Safaai[1,2,3†], Gioia De Franceschi[1§], Federica Bianca Rosselli[1¶], Walter Vanzella[1], Margherita Riggi[1], Federica Buffolo[1**], Stefano Panzeri[2], Davide Zoccolan[1*]

[1]Visual Neuroscience Lab, International School for Advanced Studies (SISSA), Trieste, Italy; [2]Laboratory of Neural Computation, Center for Neuroscience and Cognitive Systems @UniTn, Istituto Italiano di Tecnologia, Rovereto, Italy; [3]Department of Neurobiology, Harvard Medical School, Boston, United States

*For correspondence: zoccolan@sissa.it

[†]These authors contributed equally to this work

Present address: [‡]Princeton Neuroscience Institute, Princeton University, NewJersey, United States; [§]Department of Experimental Psychology, Institute of Behavioural Neuroscience, University College London, London, United Kingdom; [¶]Department of Behavior and Brain Organization, Center of Advanced European Studies and Research (Caesar), An Institute of the Max-Planck Society, Bonn, Germany; [**]Neuroscience and Brain Technologies department, Istituto Italiano di Tecnologia, Genova, Italy

**Competing interests:** The authors declare that no competing interests exist.

**Abstract** Rodents are emerging as increasingly popular models of visual functions. Yet, evidence that rodent visual cortex is capable of advanced visual processing, such as object recognition, is limited. Here we investigate how neurons located along the progression of extrastriate areas that, in the rat brain, run laterally to primary visual cortex, encode object information. We found a progressive functional specialization of neural responses along these areas, with: (1) a sharp reduction of the amount of low-level, energy-related visual information encoded by neuronal firing; and (2) a substantial increase in the ability of both single neurons and neuronal populations to support discrimination of visual objects under identity-preserving transformations (e.g., position and size changes). These findings strongly argue for the existence of a rat object-processing pathway, and point to the rodents as promising models to dissect the neuronal circuitry underlying transformation-tolerant recognition of visual objects.

## Introduction

Converging evidence (*Wang and Burkhalter, 2007*; *Wang et al., 2011*, *2012*) indicates that rodent visual cortex is organized in two clusters of strongly reciprocally connected areas, which resemble, anatomically, the primate ventral and dorsal streams (i.e., the cortical pathways specialized for the processing of, respectively, shape and motion information). The first cluster includes most of lateral extrastriate areas, while the second encompasses medial and parietal extrastriate cortex. Solid causal evidence confirms the involvement of these modules in ventral-like and dorsal-like computations – lesioning laterotemporal and posterior parietal cortex strongly impairs, respectively, visual pattern discrimination and visuospatial perception (*Gallardo et al., 1979*; *McDaniel et al., 1982*; *Wörtwein et al., 1993*; *Aggleton et al., 1997*; *Sánchez et al., 1997*; *Tees, 1999*). By comparison, functional understanding of visual processing in rodent extrastriate cortex is still limited. While studies employing parametric visual stimuli (e.g., drifting gratings) support the specialization of dorsal areas for motion processing (*Andermann et al., 2011*; *Marshel et al., 2011*; *Juavinett and Callaway, 2015*), the functional signature of ventral-like computations has yet to be found in lateral areas. In fact, parametric stimuli do not allow probing the core property of a ventral-like pathway – i.e., the ability to support recognition of visual objects despite variation in their appearance, resulting from (e.g.) position and size changes. In primates, this function, known as *transformation-tolerant* (or *invariant*) recognition, is mediated by the gradual reformatting of object representations that takes place along the ventral stream (*DiCarlo et al., 2012*). Recently, a number of behavioral studies have

**eLife digest** Everyday, we see thousands of different objects with many different shapes, colors, sizes and textures. Even an individual object – for example, a face – can present us with a virtually infinite number of different images, depending on from where we view it. In spite of this extraordinary variability, our brain can recognize objects in a fraction of a second and without any apparent effort.

Our closest relatives in the animal kingdom, the non-human primates, share our ability to effortlessly recognize objects. For many decades, they have served as invaluable models to investigate the circuits of neurons in the brain that underlie object recognition. In recent years, mice and rats have also emerged as useful models for studying some aspects of vision. However, it was not clear whether these rodents' brains could also perform complex visual processes like recognizing objects.

Tafazoli, Safaai et al. have now recorded the responses of visual neurons in rats to a set of objects, each presented across a range of positions, sizes, rotations and brightness levels. Applying computational and mathematical tools to these responses revealed that visual information progresses through a number of brain regions. The identity of the visual objects is gradually extracted as the information travels along this pathway, in a way that becomes more and more robust to changes in how the object appears.

Overall, Tafazoli, Safaai et al. suggest that rodents share with primates some of the key computations that underlie the recognition of visual objects. Therefore, the powerful sets of experimental approaches that can be used to study rats and mice – for example, genetic and molecular tools – could now be used to study the circuits of neurons that enable object recognition. Gaining a better understanding of such circuits can, in turn, inspire the design of more powerful artificial vision systems and help to develop visual prosthetics. Achieving these goals will require further work to understand how different classes of neurons in different brain regions interact as rodents perform complex visual discrimination tasks.

shown that rats too are capable of invariant recognition (*Zoccolan et al., 2009*; *Tafazoli et al., 2012*; *Vermaercke and Op de Beeck, 2012*; *Alemi-Neissi et al., 2013*; *Vinken et al., 2014*; *Rosselli et al., 2015*), thus arguing for the existence of cortical machinery supporting this function also in rodents. This hypothesis is further supported by the preferential reliance of rodents on vision during spatial navigation (*Cushman et al., 2013*; *Zoccolan, 2015*), and by the strong dependence of head-directional tuning on visual cues in rat hippocampus (*Acharya et al., 2016*). Yet, in spite of a recent attempt at investigating rat visual areas with shape stimuli (*Vermaercke et al., 2014*), functional evidence about how rodent visual cortex may support transformation-tolerant recognition is still sparse.

In our study, we compared how visual object information is processed along the anatomical progression of extrastriate areas that, in the rat brain, run laterally to V1: lateromedial (LM), laterointermediate (LI) and laterolateral (LL) areas (*Espinoza and Thomas, 1983*; *Montero, 1993*; *Vermaercke et al., 2014*). By applying specially designed information theoretic and decoding analyses, we found a sharp reduction of the amount of low-level information encoded by neuronal firing along this progression, and a concomitant increase in the ability of neuronal representations to support invariant recognition. Taken together, these findings provide compelling evidence about the existence of a ventral-like, object-processing pathway in rat visual cortex.

## Results

We used 32-channel silicon probes to record from rat primary visual cortex and the three lateral extrastriate areas LM, LI and LL (*Figure 1A* and *Figure 1—figure supplement 1*). Our experimental design was inspired by a well-established approach in ventral stream studies, consisting in passively presenting an animal (either awake or anesthetized) with a large number of visual objects in rapid sequence. This allows probing how object information is processed by the initial, largely feedforward cascade of ventral-stream computations underlying rapid recognition of visual objects

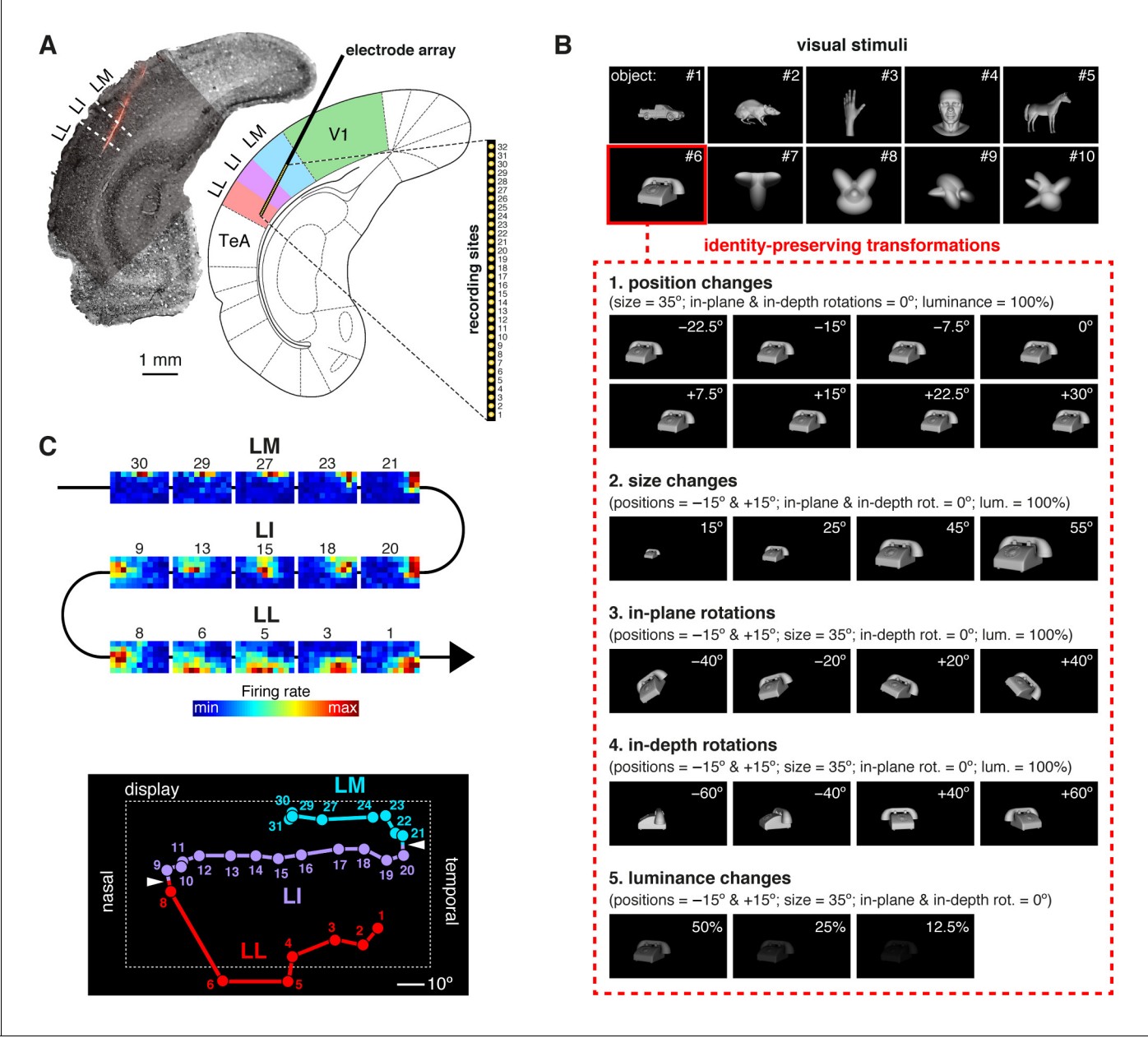

**Figure 1.** Experimental design. (**A**) Oblique insertion of a single-shank silicon probe in a typical recording session targeting rat lateral extrastriate areas LM, LI and LL, located between V1 and temporal association cortex (TeA). The probe contained 32 recording sites, spanning 1550 μm, from tip (site 1) to base (site 32). The probe location was reconstructed postmortem (left), by superimposing a bright-field image of the Nissl-stained coronal section at the targeted bregma (light gray) with an image (dark gray) showing the staining with the fluorescent dye (red), used to coat the probe before insertion (see *Figure 1—figure supplement 1*). (**B**) The stimulus set, consisting of ten visual objects (top) and their transformations (bottom). (**C**) Firing intensity maps (top) displaying the RFs recorded along the probe shown in (**A**). The numbers identify the sites each unit was recorded from. Tracking the retinotopy of the RF centers (bottom: colored dots) and its reversals (white arrows) allowed identifying the area each unit was recorded from. Details about how the stimuli were presented and the RFs were mapped are provided in *Figure 1—figure supplement 2*. The number of neurons per area obtained in each recording session (i.e., from each rat) is reported in *Figure 1—source data 1*.

The following source data and figure supplements are available for figure 1:

**Source data 1.** Number of neurons per area obtained from each rat.

**Figure supplement 1.** Histological reconstruction of the laminar location of the recording sites.

*Figure 1 continued*

**Figure supplement 2.** Computation of the receptive field size and receptive field luminance, and illustration of the tangent screen projection.

(*Thorpe et al., 1996*; *Fabre-Thorpe et al., 1998*; *Rousselet et al., 2002*) – a reflexive, stimulus-driven process that is largely independent from top-down signals and whether the animal is engaged in a recognition task (*DiCarlo et al., 2012*).

During a recording session, a rat was presented with a rich battery of visual stimuli, consisting of 10 visual objects (*Figure 1B*, top), each transformed along five variation axes (*Figure 1B*, bottom), for a total of 380 stimulus conditions (i.e., object views). These conditions included combinations of object identities (i.e., objects #7–10) and transformations that we have previously shown to be invariantly recognized by rats (*Tafazoli et al., 2012*). In addition, drifting bars were used to map the receptive field (RF) of each recorded unit (*Figure 1—figure supplement 2A–B*). To allow for the repeated presentation (average of 26.5 trials per stimulus) of such large number of stimulus conditions, while maintaining a good stability of the recordings, rats were kept in an anesthetized state during each session (see Discussion for possible implications).

We recorded 771 visually driven and stimulus informative units in 26 rats: 228 from V1, 131 from LM, 260 from LI, and 152 from LL (neurons in each area came from at least eight rats; *Figure 1—source data 1*). With 'units' we refer here to a combination of both well-isolated single units and multiunit clusters. Hereafter, most results will be presented over the whole population of units, although a number of control analyses were carried out on well-isolated single units only (see Discussion). The cortical area each unit was recorded from was identified by tracking the progression of the RFs recorded along a probe (*Figure 1C*), so as to map the reversals of the retinotopy that, in rodent visual cortex, delineate the borders between adjacent areas (*Schuett et al., 2002*; *Wang and Burkhalter, 2007*; *Andermann et al., 2011*; *Marshel et al., 2011*; *Polack and Contreras, 2012*; *Vermaercke et al., 2014*). This procedure was combined with the histological reconstruction of the probe insertion track (*Figure 1A*) and, when possible, of the laminar location of the individual recording sites (*Figure 1—figure supplement 1*).

## The fraction of energy-independent stimulus information sharply increases from V1 to LL

Under the hypothesis that, along an object-processing pathway, information about low-level image properties should be partially lost (*DiCarlo et al., 2012*), we measured the sensitivity of each recorded neuron to the lowest-level attribute of the visual input – the amount of luminous energy a stimulus impinges on a neuronal receptive field. This was quantified by a metric (*RF luminance*; see Materials and methods), which, for some neuron, seemed to account for the modulation of the firing rate not only across object transformations, but also across object identities. This was the case for the example V1 neuron shown in *Figure 2A*, where the objects eliciting the larger responses along the position axis (red and blue curves) were the ones that consistently covered larger fractions of the neuron's RF (yellow ellipses), thus yielding higher RF luminance values (*Figure 2—figure supplement 1A–B*), as compared to the less effective object (green curve). By contrast, for other units (e. g., the example LL neuron shown in *Figure 2B*), RF luminance did not appear to account for the tuning for object identity – similarly bright objects (*Figure 2—figure supplement 1C–D*) yielded very different response magnitudes (compare the red with the green and blue curves).

To better appreciate the sensitivity of each neuron to stimulus energy, we considered a subset of the stimulus conditions, consisting of 23 transformations of each object, for a total of 230 stimuli (see Materials and methods). We then grouped these stimuli in 23 equi-populated RF luminance bins and we color-coded the intensity of the neuronal response across the resulting matrix of 10 stimuli per 23 bins (the stimuli within each bin were ranked according to the magnitude of the response they evoked). Many units, as the example V1 neuron of *Figure 2C* (same cell of *Figure 2A*), showed a gradual increase of activity across consecutive bins of progressively larger luminance, with little variation of firing within each bin. Other units, as the example LL neuron of *Figure 2D* (same cell of *Figure 2B*), displayed no systematic variation of firing across the luminance axis, but were strongly

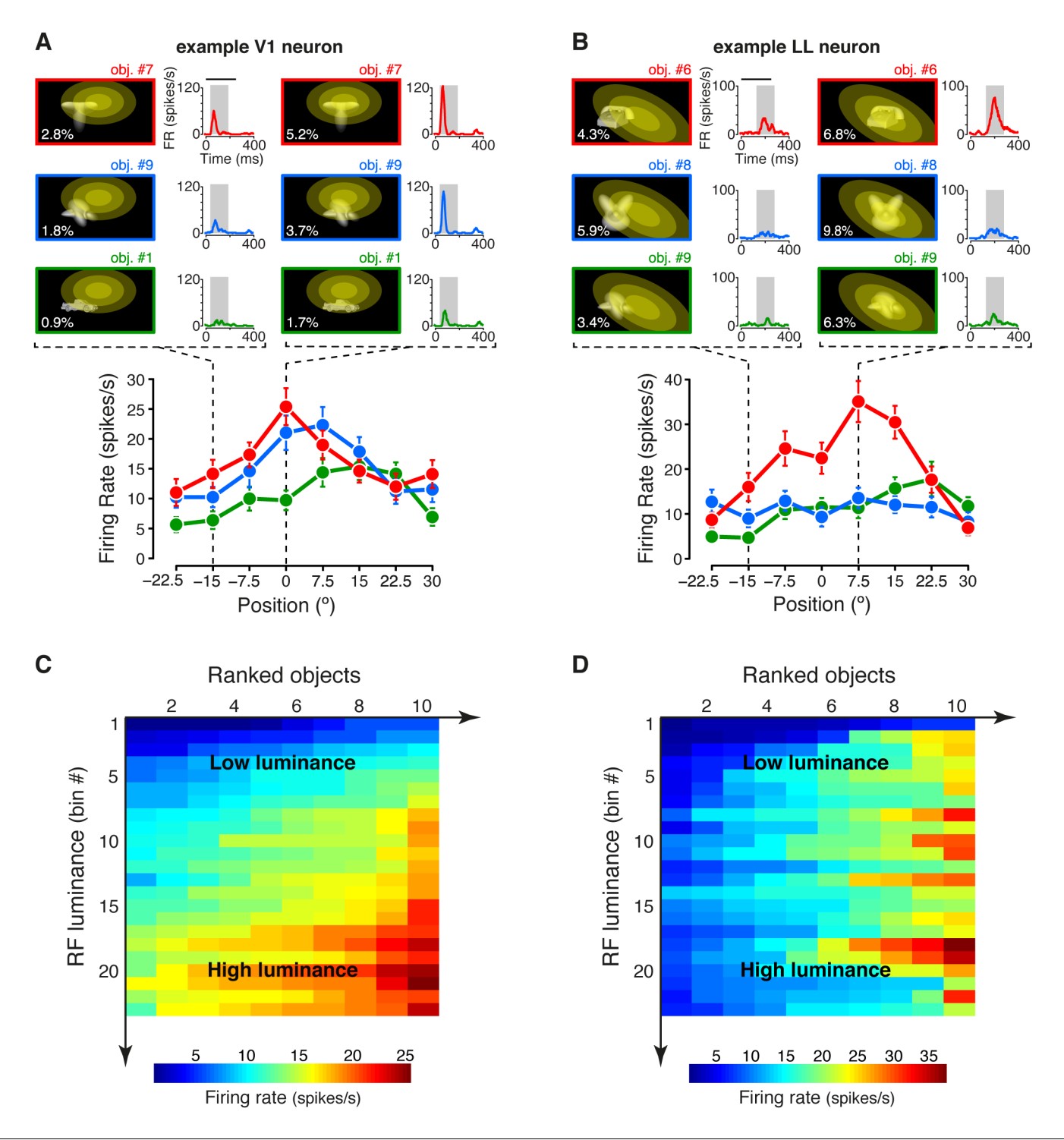

**Figure 2.** Tuning of an example V1 and LL neuron. (A–B) The bottom plots show the average firing rates (AFRs) of a V1 (A) and a LL (B) neuron, evoked by three objects, presented at eight different visual field positions (shown in *Figure 1B.1*). The top plots show the peri-stimulus time histograms (PSTHs) obtained at two positions, along with the images of the corresponding stimulus conditions. These images also display the RF profile of each neuron, in the guise of three concentric ellipses, corresponding to 1 SD, 2 SD and 3 SD of the two-dimensional Gaussians that were fitted to the raw RFs. The numbers (white font) show the luminosity that each object condition impinged on the RF (referred to as *RF luminance*; the distributions of RF luminance values produced by the objects across the full set of positions and the full set of transformations are shown in *Figure 2—figure supplement 1*). The gray patches over the PSTHs show the spike count windows (150 ms) used to compute the AFRs (their onsets were the response latencies of the

*Figure 2 continued on next page*

*Figure 2 continued*

neurons). Error bars are SEM. (**C–D**) Luminance sensitivity profiles for the two examples neurons shown in (**A**) and (**B**). For each neuron, the stimulus conditions were grouped in 23 RF luminance bins with 10 stimuli each, and the intensity of firing across the resulting 23 × 10 matrix was color-coded. Within each bin, the stimuli were ranked according to the magnitude of the response they evoked.

The following figure supplement is available for figure 2:

**Figure supplement 1.** RF luminance values produced by very effective and poorly effective objects: a comparison between an example V1 and an example LL neuron.

modulated by the conditions within each luminance bin, thus suggesting a tuning for higher-level stimulus properties.

Note that, although the example LL neuron of *Figure 2D* fired more sparsely than the example V1 neuron of *Figure 2C*, the sparseness of neuronal firing across the 230 stimulus conditions, measured as defined in (*Vinje and Gallant, 2000*), was not statistically different between the LL and the V1 populations (p>0.05; Mann-Whitney U-test), with the median sparseness being ~0.13 in both areas. Critically, this does not imply that the two areas do not differ in the way they encode visual objects, because sparseness is a combined measure of object selectivity and tolerance to changes in object appearance, which is positively correlated with the former and negatively correlated with the latter. As such, a concomitant increase of both selectivity and tolerance can lead to no appreciable change of sparseness across an object-processing hierarchy (*Rust and DiCarlo, 2012*). This suggests that other approaches are necessary to compare visual object representations along a putative ventral-like pathway.

In our study, we first quantified the relative sensitivity of a neuron to stimulus luminance and higher-level features by using information theory, because of two main advantages this approach offers in investigating neuronal coding. First, computing mutual information between a stimulus' feature and the evoked neuronal response provides an upper bound to how well we can reconstruct the stimulus' feature from observing the neural response on a single trial, without committing to the choice of any specific decoding algorithm (*Rieke et al., 1997*; *Borst and Theunissen, 1999*; *Quiroga and Panzeri, 2009*; *Rolls and Treves, 2011*). Second, information theory provides a solid mathematical framework to disentangle the ability of a neuron to encode a given stimulus' feature from its ability to encode another feature, even if these two features are not independently distributed across the stimuli, i.e., even if they are correlated in an arbitrarily complex, non-linear way (*Ince et al., 2012*). This property was crucial to allow estimating the relative contribution of luminance and higher-level features to the tuning of rat visual neurons, given that luminance did co-vary, in general, with other stimulus properties, such as object identity, position, size, etc. (see *Figure 2A–B*).

In our analysis, for each neuron, we first computed Shannon's mutual information between stimulus identity $S$ and neuronal response $R$, formulated as:

$$I(R;S) = \sum_{s} P(s) \sum_{r} P(r|s) \log_2 \frac{P(r|s)}{P(r)}, \tag{1}$$

where *P(s)* is the probability of presentation of stimulus *s*, *P(r|s)* is the probability of observing a response *r* following presentation of stimulus *s*, and *P(r)* is the probability of observing a response *r* across all stimulus presentations. The response *R* was quantified as the number of spikes fired by the neuron in a 150 ms-wide spike count window (e.g., see the gray patches in *Figure 2A–B*), while the stimulus conditions *S* included all the 23 transformations of the 10 objects, previously used to produce *Figure 2C–D*, for a total of 230 different stimuli (see Materials and methods for details). As graphically illustrated in *Figure 3A*, $I(R;S)$ measured the discriminability of these 230 different stimuli, given the spike count distributions they evoked in a single neuron.

We then decomposed this overall stimulus information $I(R;S)$ into the sum of the information about stimulus luminance and the information about luminance-independent, higher-level features, using the following mathematical identity (*Ince et al., 2012*):

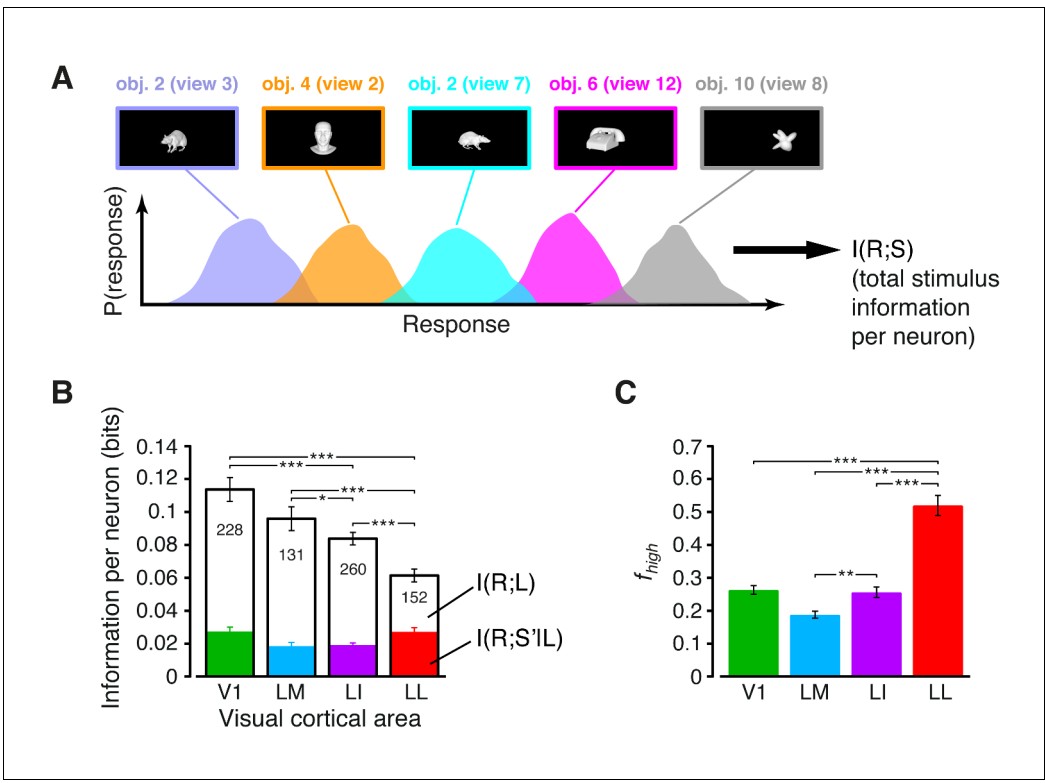

**Figure 3.** Information conveyed by the neuronal response about stimulus luminance and luminance-independent visual features. (**A**) Illustration of how total stimulus information per neuron was computed. All the views of all the objects (total of 230 stimuli) were considered as different stimulus conditions, each giving rise to its own response distribution (colored curves; only a few examples are shown here). Mutual information between stimulus and response measured the discriminability of the stimuli, given the overlap of their response distributions. (**B**) Mutual information (median over the units recorded in each area ± SE) between stimulus and response in each visual area (full bars; *p<0.05, **p<0.01, ***p<0.001; 1-tailed U-test, Holm-Bonferroni corrected). The white portion of the bars shows the median information that each area carried about stimulus luminance, while the colored portion is the median information about higher-order, luminance-independent visual features. The number of cells in each area is written on the corresponding bar. (**C**) Median fraction of luminance-independent stimulus information ($f_{high}$; see Results) that neurons carried in each area. Error bars and significance levels/test as in (**B**). The mutual information metrics obtained for neurons sampled from cortical layers II-IV and V-VI are reported in *Figure 3—figure supplement 1*. The $f_{high}$ values obtained for neuronal subpopulations with matched spike isolation quality are shown in *Figure 3—figure supplement 2*. The sensitivity of rat visual neurons to luminance variations of the same object is shown in *Figure 3—figure supplement 3*. The information carried by rat visual neurons about stimulus contrast and contrast-independent visual features is reported in *Figure 3—figure supplement 4*.

The following figure supplements are available for figure 3:

**Figure supplement 1.** Information conveyed by the neuronal response about stimulus luminance and luminance-independent visual features: a comparison between superficial and deep layers.

**Figure supplement 2.** Independence of the fraction of luminance-independent stimulus information from the quality of spike isolation.

**Figure supplement 3.** Sensitivity of rat visual neurons to luminance variations of the same object.

**Figure supplement 4.** Information conveyed by the neuronal response about stimulus contrast and contrast-independent visual features.

$$I(R;S) \equiv I(R;S'\&L) = I(R;S'|L) + I(R;L). \qquad (2)$$

Here, $L$ is the RF luminance of the visual stimuli; $I(R;L)$ is the information that $R$ conveys about $L$; and $I(R;S'|L)$ measures how much information $R$ carries about a variable $S'$ that denotes the identity of each stimulus condition $S$, as defined by any possible visual attribute with the exception of the RF luminance $L$ (i.e., $S = S' \& L$).

The overall amount of visual information decreased gradually and significantly along the areas' progression (full bars in *Figure 3B*), with the median *I(R;S)* being about half in LL (~0.06 bits) than in V1 (~0.12 bits; 1-tailed, Mann-Whitney U-test, Holm-Bonferroni corrected for multiple comparisons; hereafter, unless otherwise stated, all the between-area comparisons have been statistically assessed with this test; see Material and methods). This decline was due to a loss of the energy-related information $I(R;L)$ (white portion of the bars), rather than to a drop of the higher-level, energy-independent information $I(R;S'|L)$ (colored portion of the bars), which changed little across the areas. As a result, the fraction of total information that neurons carried about higher-level visual attributes, i.e., $f_{high} = I(R;S'|L)/I(R;S)$, became about twice as large in LL (median ~0.5) as in V1, LM and LI, and such differences were all highly significant (*Figure 3C*). All these trends were largely preserved when neurons in superficial and deep layers were considered separately (*Figure 3—figure supplement 1*) and did not depend on the quality of spike isolation (*Figure 3—figure supplement 2*; see Discussion).

The decrease of sensitivity to stimulus luminance along the areas' progression was confirmed by measuring the tuning of rat visual neurons across the four luminance changes each object underwent (i.e., 12.5%, 25%, 50% and 100% luminance; see *Figure 1B.5*). In all the areas, the luminance-sensitivity curves showed a tendency of the firing rate to increase as a function of object luminance (*Figure 3—figure supplement 3A*). However, such a growth was steeper in V1 and LM, as compared to LI and LL, where several neurons had a relatively flat tuning, with a peak, in some cases, at intermediate luminance levels. These trends were quantified by computing the sparseness of the response of each neuron over the luminance axis, which decreased monotonically along the areas' progression (*Figure 3—figure supplement 3B*), thus confirming the drop of sensitivity to luminance from V1 to LL.

To further explore whether rat lateral visual areas were differentially sensitive to other low-level properties, we defined a metric (*RF contrast*; see Materials and methods) that quantified the variability of the luminance pattern impinged by any given stimulus on a neuronal RF. We then measured how much information rat visual neurons carried about RF contrast, and how much information they carried about contrast-independent visual features (i.e., we applied *Equation 2*, but with RF contrast instead of RF luminance). The information about RF contrast decreased monotonically along the areas' progression (white portion of the bars in *Figure 3—figure supplement 4A*), while the contrast-independent information peaked in LL. As a consequence, the fraction of contrast-independent information carried by neuronal firing grew sharply and significantly from V1 to LI (*Figure 3—figure supplement 4B*). Taken together, the results presented in this section show a clear tendency for low-level visual information to be substantially pruned along rat lateral extrastriate areas.

## The amount of view-invariant object information gradually increases from V1 to LL

Next, we explored whether neurons along lateral extrastriate areas also become gradually more capable of coding the identity of visual objects in spite of variation in their appearance (*DiCarlo et al., 2012*). To this aim, we relied on both information theoretic and linear decoding analyses. Both approaches have been extensively used to investigate the primate visual system, with mutual information yielding estimates of the capability of single neurons to code both low-level (e. g., contrast and orientation) and higher-level (e.g., faces) visual features at different time resolutions and during different time epochs of the response (*Optican and Richmond, 1987*; *Tovee et al., 1994*; *Rolls and Tovee, 1995*; *Sugase et al., 1999*; *Montemurro et al., 2008*; *Ince et al., 2012*), and linear decoders probing the suitability of neuronal populations to support easy readout of object identity (*Hung et al., 2005*; *Li et al., 2009*; *Rust and Dicarlo, 2010*; *Pagan et al., 2013*; *Baldassi et al., 2013*; *Hong et al., 2016*). In our study, we used both methods because of their complementary advantages. By computing mutual information, we estimated the overall amount of

transformation-invariant information that single neurons conveyed about object identity. By applying linear decoders, we measured what fraction of such invariant information was formatted in a convenient, easy-to-read-out way, both at the level of single neurons and neuronal populations.

In the information theoretic analysis, we defined every stimulus condition $S$ as a combination of object identity $O$ and transformation $T$ (i.e., $S = O$ and $T$) and we expressed the overall stimulus information $I(R;S)$ as follows:

$$I(R;S) \equiv I(R;O\&T) = I(R;T|O) + I(R;O). \tag{3}$$

Here, $I(R;O)$ is the amount of view-invariant object information carried by neuronal firing, i.e., the information that $R$ conveys about object identity, when the responses produced by the 23 transformations of an object (across repeated presentations) are considered together, so as to give rise to an overall response distribution (see illustration in *Figure 4A*, bottom). The other term, $I(R;T|O)$, is the information that $R$ carries about the specific transformation of an object, once its identity has been fixed.

Given a neuron, we computed these information metrics for all possible pairs of object identities, while, at the same time, measuring the similarity between the objects in each pair in terms of RF luminance. Such similarity was evaluated by computing the ratio between the mean luminance of the dimmer object (across its 23 views) and the mean luminance of the brighter one – the resulting *luminosity ratio* ranged from zero (dimmer object fully dark) to one (both objects with the same luminance). By considering object pairs with a luminosity ratio larger than a given threshold $Th_{LumRatio}$, and allowing $Th_{LumRatio}$ to range from zero to one, we could restrict the computation of the information metrics to object pairs that were progressively less discriminable based on luminance differences, thus probing to what extent the ability of a neuron to code invariantly object identity depended on its luminance sensitivity.

The overall stimulus information per neuron $I(R;S)$ followed the same trend already shown in *Figure 3B* (i.e., it decreased gradually along the areas' progression) and such trend remained largely unchanged as a function of $Th_{LumRatio}$ (*Figure 4B*). By contrast, the amount of view-invariant object information per neuron $I(R;O)$ strongly depended on $Th_{LumRatio}$ (*Figure 4C*, left). When all object pairs were considered ($Th_{LumRatio} = 0$), areas LM and LI conveyed the largest amount of invariant information, followed by V1 and LL. This trend remained stable until $Th_{LumRatio}$ reached 0.3, after which the amount of invariant information in V1, LM and LI dropped sharply. By contrast, the invariant information in LL remained stable until $Th_{LumRatio}$ approached 0.7, after which it started to increase. As a result, when only pairs of objects with very similar luminosity were considered ($Th_{LumRatio} = 0.9$), a clear gradient emerged across the four areas, with the invariant information in LI and LL being significantly larger than in V1 and LM (*Figure 4C*, right).

These results indicate that neurons in V1, LM and LI were able to rely on their sharp sensitivity to luminous energy (*Figure 3B*) and use luminance as a cue to convey relatively large amount of invariant object information, when such cue was available (i.e., for small values of $Th_{LumRatio}$). The example V1 neuron shown in *Figure 2A* is one of such units. This cell successfully supported the discrimination of some object pairs only because of its strong sensitivity to stimulus luminance (*Figure 2C*), which, for those pairs, happened to co-vary with object identity, in spite of the position changes and the other transformations that the objects underwent (*Figure 2—figure supplement 1A–B*). In cases like this, observing a large amount of view-invariant information would be an artifact, because luminance would not at all be diagnostic of object identity, if these neurons were probed with a variety of object appearances as large as the one experienced during natural vision (where each object can project thousands of different images on the retina). It is only because of the limited range of transformations that are testable in a neurophysiology experiment that luminance can possibly serve as a transformation-invariant cue of object identity.

To verify that luminance could indeed act as a transformation-invariant cue, we measured $I(L;O)$ – the amount of view-invariant object information conveyed by RF luminance alone (*Figure 4—figure supplement 1*). As expected, when no restriction was applied to the luminance difference of the objects to discriminate (i.e., for small values of $Th_{LumRatio}$), $I(L;O)$ was very large in all the areas. This confirmed that RF luminance, by itself, was able to convey a substantial amount of invariant object information. When $Th_{LumRatio}$ was allowed to increase, $I(L;O)$ dropped sharply, eventually reaching zero in all the areas for $Th_{LumRatio} = 0.9$. Interestingly, this decrease was the same found for $I(R;O)$ in

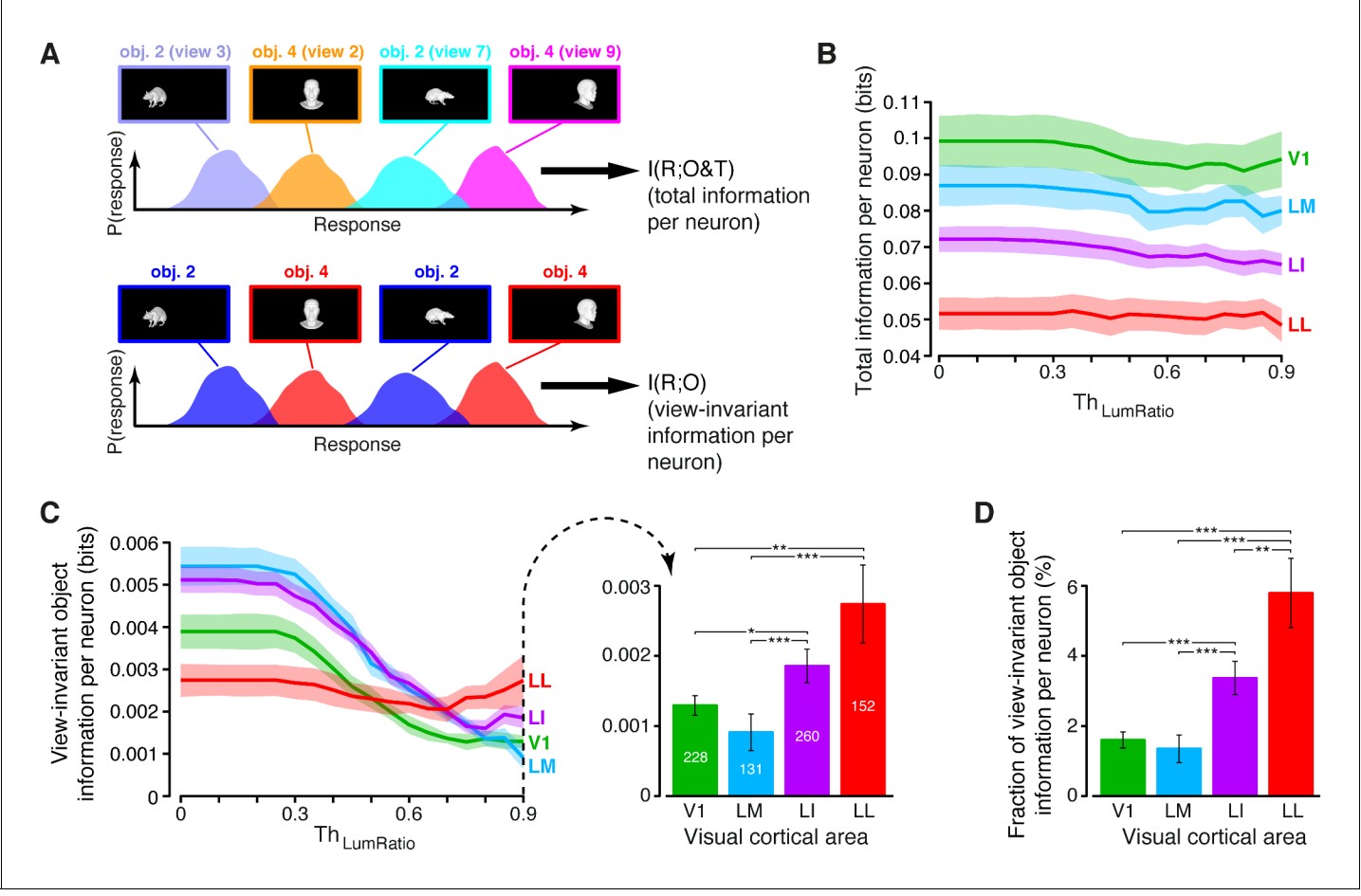

**Figure 4.** Comparing total visual information and view-invariant object information per neuron. (**A**) Illustration of how total visual information and view-invariant object information per neuron were computed, given an object pair. In the first case, all the views of the two objects were considered as different stimulus conditions, each giving rise to its own response distribution (colored curves). In the second case, the response distributions produced by different views of the same object were merged into a single, overall distribution (shown in blue and red, respectively, for the two objects). (**B**) Total visual information per neuron (median over the units recorded in each area ± SE) as a function of the similarity between the RF luminance of the objects in each pair, as defined by $Th_{LumRatio}$ (see Results). (**C**) Left: view-invariant object information per neuron (median ± SE) as a function of $Th_{LumRatio}$. Right: view-invariant object information per neuron (median ± SE) for $Th_{LumRatio} = 0.9$ (*p<0.05, **p<0.01, ***p<0.001; 1-tailed U-test, Holm-Bonferroni corrected). The number of cells in each area is written on the corresponding bar. (**D**) Ratio between view-invariant object information and total information per neuron (median ± SE), for $Th_{LumRatio} = 0.9$. Significance levels/test as in (**C**). The invariant object information carried by RF luminance as a function of $Th_{LumRatio}$ is reported in *Figure 4—figure supplement 1*.

The following figure supplement is available for figure 4:

**Figure supplement 1.** Invariant object information carried by RF luminance as a function of $Th_{LumRatio}$.

V1, LM and LI (see *Figure 4C*), thus showing that a large fraction of the invariant information observed in these areas (but not in LL) at low $Th_{LumRatio}$ was indeed accounted for by luminance differences between the objects in the pairs. Hence, the need of setting $Th_{LumRatio} = 0.9$, thus considering only pairs of objects with very similar luminance to nullify the luminance confound, when comparing the areas in terms of their ability to support invariant recognition.

Our analysis shows that, when this restriction was applied, a clear gradient emerged along the areas' progression, with LL conveying the largest amount of invariant information per neuron, followed by LI and then by V1/LM (*Figure 4C*, right). A similar, but sharper trend was observed when the relative contribution of the view-invariant information to the total information was measured, i. e., when the ratio between $I(R;O)$ and $I(R;S)$ at $Th_{LumRatio} = 0.9$ was computed (*Figure 4D*). The

fraction of invariant information increased very steeply and significantly along the areas' progression, being almost four times larger in LL than in V1/LM, and ~1.7 times larger in LL than in LI. Overall, these results indicate that the information that single neurons are able to convey about the identity of visual objects, in spite of variation in their appearance, becomes gradually larger along rat lateral visual areas (see Discussion for further implications of these findings).

## Object representations become more linearly separable from V1 to LL, and better capable of supporting generalization to novel object views

$I(R;O)$ (*Figure 4C*) provides an upper bound to the amount of transformation-invariant information that single neurons can encode about object identity, but it does not quantifies how easily this information can be read out by simple linear decoders (see illustration in *Figure 5A*). Assessing this property, known as *linear separability* of object representations, is crucial, because attaining progressively larger levels of linear separability is considered the key computational goal of the ventral stream (*DiCarlo et al., 2012*). In our study, we first measured linear separability at the single-cell level – i.e., given a neuron and a pair of objects, we tested the ability of a binary linear decoder to correctly label the responses produced by the 23 views of each object (this was done using the cross-validation procedure described in Materials and methods). For consistency with the previous analyses, the discrimination performance of the decoders was computed as the mutual information between the actual and the predicted object labels from the decoding outcomes (*Quiroga and Panzeri, 2009*).

The linear separability of object representations at the single-neuron level increased monotonically and significantly along the areas' progression (*Figure 5B*), being ~4 times larger in LL as in V1 and LM, and reaching an intermediate value in LI. This increase was steeper than the growth of the

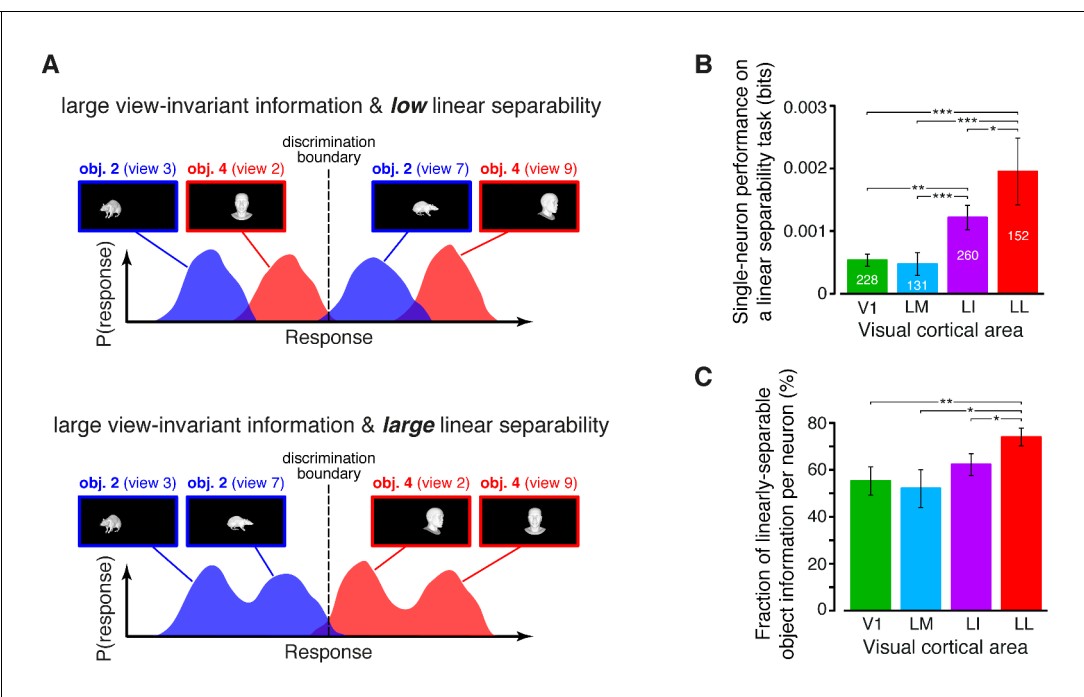

**Figure 5.** Linear separability of object representations at the single-neuron level. (A) Illustration of how a similar amount of view-invariant information per neuron can be encoded by object representations with a different degree of linear separability. The overlap between the response distributions produced by two objects across multiple views is similarly small in the top and bottom examples; hence, the view-invariant object information per neuron is similarly large. However, only for the distributions shown at the bottom, a single discrimination boundary can be found (dashed line) that allows discriminating the objects regardless of their specific view. (B) Linear separability of object representations at the single-neuron level (median over the units recorded in each area ± SE; *p<0.05, **p<0.01, ***p<0.001; 1-tailed U-test, Holm-Bonferroni corrected). The number of cells in each area is written on the corresponding bar. (C) Ratio between linear separability, as computed in (B), and view-invariant object information, as computed in *Figure 4C*, right (median ± SE). Significance levels/test as in (B). In both (B) and (C), $Th_{LumRatio} = 0.9$.

view-invariant information $I(R;O)$ (*Figure 4C*, right), thus suggesting that not only LL neurons encoded more invariant information than neurons in the other areas, but also that a lager fraction of this information was linearly decodable in LL. To quantify this observation we computed, for each cell, the ratio between linear separability (*Figure 5B*) and amount of invariant information (*Figure 4C*, right). The resulting fraction of invariant information that was linearly decodable increased from ~55% in V1 to ~70% in LL, being significantly larger in LL than in any other area (*Figure 5C*).

An alternative (more stringent) way to assess transformation-tolerance is to measure to what extent a linear decoder, trained with a single view per object, is able to discriminate the other (untrained) views of the same objects (see illustration in *Figure 6A*) – a property known as *generalization* across transformations. Since, at the population level, large linear separability does not necessarily imply large generalization performance (*Rust and Dicarlo, 2010*) (see also Figure 8A), it was important to test rat visual neurons with regard to the latter property too (see Materials and methods). Our analysis revealed that, when assessed at the single-cell level, the ability of rat visual neurons to support generalization to novel object views increased as significantly and steeply as linear separability (*Figure 6B*). Interestingly, this trend was widespread across the whole cortical thickness and equally sharp in superficial and deep layers, although the generalization performances were larger in the latter (*Figure 6C*). These conclusions were supported by a two-way ANOVA with *visual area* and *layer* as factors, yielding a significant main effect for both area and layer (p<0.001, $F_{3,687} = 9.9$ and $F_{1,687} = 6.93$, respectively) but no significant interaction (p>0.15, $F_{3,687} = 1.74$).

The growth of transformation tolerance from V1 to LL was also observed when the individual transformation axes were considered separately in the decoding analysis (*Figure 6D*). In this case, the training and testing views of the objects to be decoded were all randomly sampled from the same axis of variation: either position, size, rotation (both in-plane and in-depth, pooled together) or luminance. In all four cases, LL yielded the largest decoding performance, typically followed by LI and then by the more medial areas (V1 and LM). The difference between LL and V1/LM was statistically significant for the position, size and rotation changes. Other comparisons were also significant, such as LL vs. LI for the position and size transformations, and LI vs. V1 and LM for the rotations.

To further compare the position tolerance afforded by the four visual areas, we also measured the ability of single neurons to support generalization across increasingly wider positions changes – a decoder was trained to discriminate two objects presented at the same position in the visual field, and then tested for its ability to discriminate those same objects, when presented at positions that were increasingly distant from the training location (Materials and methods). The neurons in LL consistently yielded the largest generalization performance, which remained very stable (invariant) as a function of the distance from the training location (*Figure 6—figure supplement 1A*). A much lower performance was observed for LI and LM, while V1 displayed the steepest decrease along the distance axis (these observations were all statistically significant, as assessed by a two-way ANOVA with *visual area* and *distance from the training position* as factors; see the legend of *Figure 6—figure supplement 1A* for details). A similar analysis was carried out to quantify the tolerance to size changes. In this case, we trained binary decoders to discriminate two objects at a given size, and then tested how well they generalized when those same objects were shown at either smaller or larger sizes (Materials and methods). The resulting patterns of generalization performances (*Figure 6—figure supplement 1B*) confirmed once more the significantly larger tolerance afforded by LL neurons, as compared to the other visual areas. Taken together, the results of *Figure 6D* and *Figure 6—figure supplement 1* indicate that the growth of tolerance across the areas' progression was widespread across all tested transformations axes, ultimately yielding to the emergence, in LL, of a general-purpose representation that tolerated a wide spectrum of image-level variation (see Discussion).

Inspired by previous studies of the ventral stream (*Li et al., 2009*; *Rust and Dicarlo, 2010*), we also tested to what extent RF size played a role in determining the growth of invariance across rat lateral visual areas (intuitively, neurons with large RFs should respond to their preferred visual features over a wide span of the visual field, thus displaying high position and size tolerance). Earlier investigations of rat visual cortex have shown that RF size increases along the V1-LM-LI-LL progression (*Espinoza and Thomas, 1983*; *Vermaercke et al., 2014*). Our recordings confirmed and expanded these previous observations, showing that RF size (defined in Materials and methods) grew significantly at each step along the areas' progression (*Figure 7A*), with the median in LL (~30°

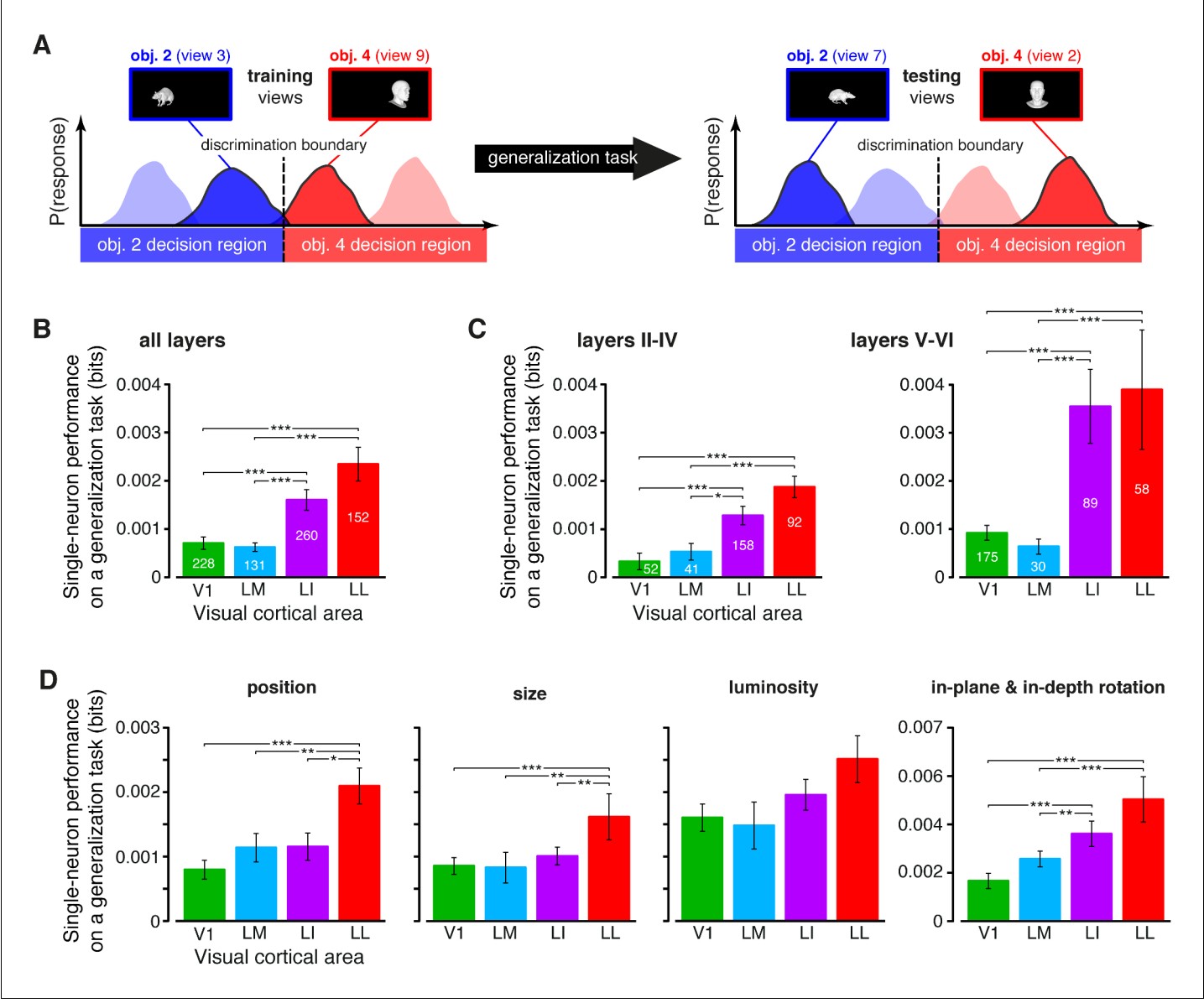

**Figure 6.** Ability of single neurons to support generalization to novel object views. (**A**) Illustration of how the ability of single neurons to discriminate novel views of previously trained objects was assessed. The blue and red curves refer to the hypothetical response distributions evoked by different views of two objects. A binary decoder is trained to discriminate two specific views (darker curves, left), and then tested for its ability to correctly recognize two different views (darker curves, right), using the previously learned discrimination boundary (dashed lines). (**B**) Generalization performance achieved by single neurons with novel object views (median over the units recorded in each area ± SE; *p<0.05, **p<0.01, ***p<0.001; 1-tailed U-test, Holm-Bonferroni corrected). The number of cells in each area is written on the corresponding bar. (**C**) Median generalization performances (± SE) achieved by single neurons for the neuronal subpopulations sampled from cortical layers II-IV (left) and V-VI (right). Significance levels/test as in (**B**). Note that the laminar location was not retrieved for all the recorded units (see Materials and methods). (**D**) Median generalization performances (± SE) achieved by single neurons, computed along individual transformation axes. In (**C–D**), significance levels/test are as in (**B**). $Th_{LumRatio} = 0.9$ in (**B–D**). The generalization performances achieved by single neurons across parametric position and size changes are reported in *Figure 6—figure supplement 1*. The generalization performances obtained for neuronal subpopulations with matched spike isolation quality are shown in *Figure 6—figure supplement 2*. The firing rate magnitude measured in the four areas (before and after matching the neuronal populations in terms of spike isolation quality) is reported in *Figure 6—figure supplement 3*.

The following figure supplements are available for figure 6:

**Figure supplement 1.** Generalization achieved by single neurons across parametric position and size changes.

**Figure supplement 2.** Independence of the generalization performances yielded by single neurons from the quality of spike isolation.

*Figure 6 continued on next page*

*Figure 6 continued*

**Figure supplement 3.** Magnitude of the firing rates.

of visual angle) being about twice as large as in V1. At the same time, RF size varied widely within each area, resulting in a large overlap among the distributions obtained for the four populations. This allowed sampling the largest possible V1, LM, LI and LL subpopulations with matched RF size ranges (gray patches in *Figure 7B*), and computing the generalization performances yielded by these subpopulations (*Figure 7C*) – again, LL and LI afforded significantly larger tolerance than V1 and LM. A similar result was found for the fraction of energy-independent stimulus information $f_{high}$ carried by these RF-matched subpopulations (*Figure 7D*) – $f_{high}$ followed the same trend observed for the whole populations (*Figure 3C*), with a sharp, significant increase from the more medial areas to LL.

Interestingly, the decoding performances obtained for the RF-matched subpopulations were larger than those obtained for the whole populations, especially in the more lateral areas (compare *Figure 7C* to *Figure 6B*). This means that the range of RF sizes that was common to the four populations contained the neurons that, in each area, yielded the largest decoding performances. This was the result of the specific relationship that was observed, within each neuronal population, between performance and RF size (*Figure 7B*). While in V1 performance slightly increased as a function of RF size (although not significantly; p>0.05; two-tailed t-test), performance and RF size were significantly anti-correlated in LI and LL (p<0.01). As explained in the Discussion, these findings suggest a striking similarity with the tuning properties of neurons at the highest stages of the monkey ventral stream (*DiCarlo et al., 2012*).

## Linear readout of population activity confirms the steep increase of transformation-tolerance along rat lateral visual areas

The growth of transformation tolerance reported in the previous section was highly significant and quite substantial, in relative terms. In LL, the discrimination performances were typically 2–4 times larger than in V1, yet, in absolute terms, their magnitude was in the order of a few thousandths of a bit. This raised the question of whether such single-unit variations would translate into macroscopic differences among the areas, at the neuronal population level.

To address this issue, we performed a population decoding analysis, in which we trained binary linear classifiers to read out visual object identity from the activity of neuronal populations of increasing size in V1, LM, LI and LL. Random subpopulations of $N$ units, with $N = \{6, 12, 24, 48, 96\}$, were sampled from the full sets of neurons in each area. Given a subpopulation, the response axes of the sampled units formed a vector space, where each object condition was represented by the cloud of population response vectors produced by the repeated presentation of the condition across multiple trials (see illustration in *Figure 8A*). The binary classifiers were required to correctly label the population vectors produced by different pairs of objects across transformations, thus testing to what extent the underlying object representations were linearly separable (*Figure 8A*, left) and generalizable (*Figure 8A*, right). To avoid the luminance tuning confound, a pair of objects was included in the analysis, only if at least 96 neurons in each area could be found for which the luminance ratio of the pair was larger than a given threshold $Th_{LumRatio}$. We set $Th_{LumRatio}$ to the largest value (0.8) that yielded at least three object pairs (*Figure 8B* show the pairs that met this criterion, while *Figure 8C and E* show the mean classification performances over these three pairs).

The ability of the classifiers to linearly discriminate the object pairs increased sharply as a function of $N$ (*Figure 8C*, left). In LL, the performance grew of ~500%, when $N$ increased from 6 to 96, reaching ~0.22 bits, which was nearly twice as large as the performance obtained in LI. More in general, linear separability grew considerably along the areas' progression, following the same trend observed at the single-cell level (see *Figure 5B*), but with performances that were about two orders of magnitude larger (for $N = 96$). As a result, the differences among the areas became macroscopic – when measured in terms of classification accuracy (*Figure 8C*, right), LL performance (~76% correct discrimination) was about eight percentage points above LI, 10 above V1 and 17 above LM. Given

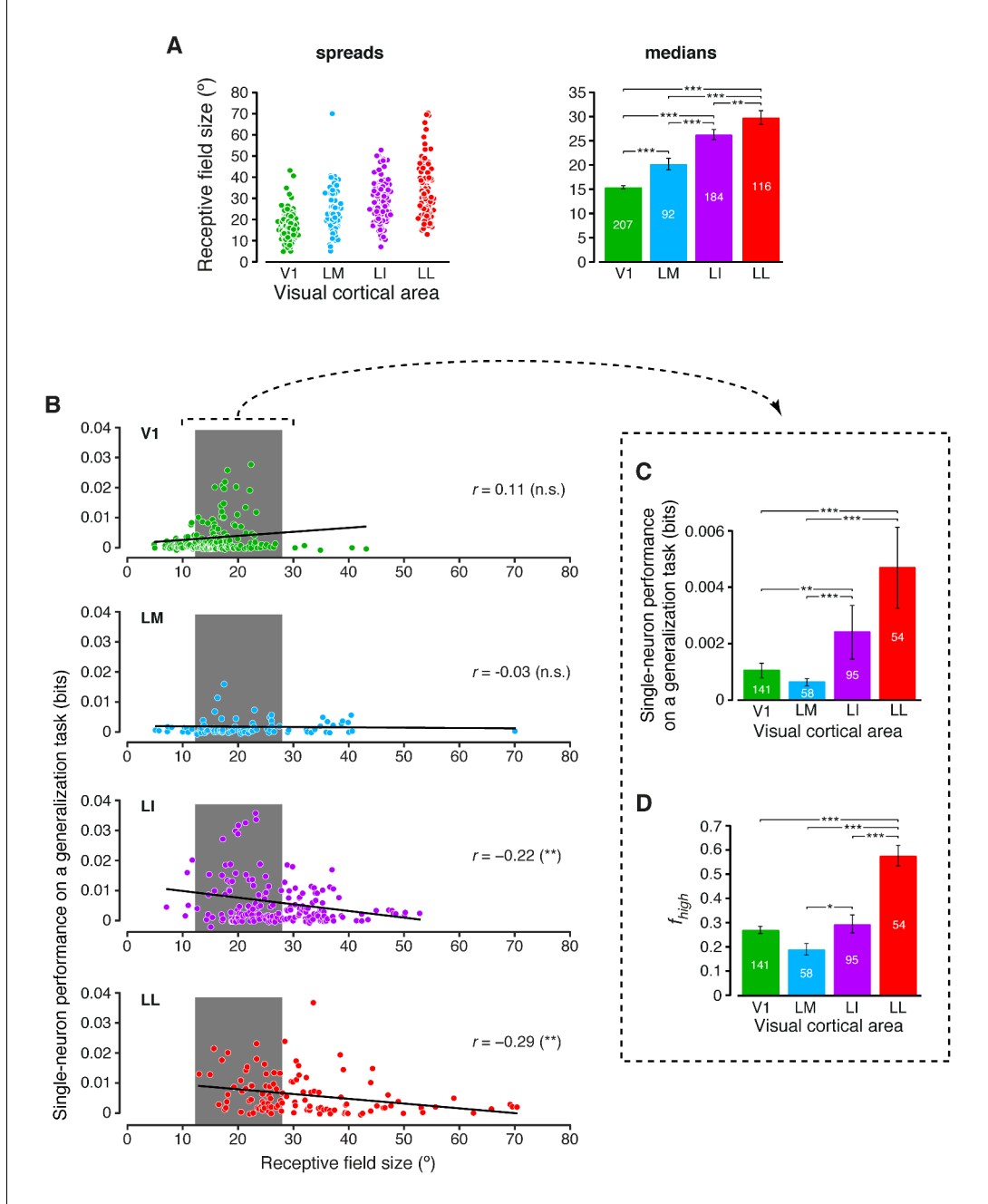

**Figure 7.** Single-neuron decoding and mutual information metrics, compared across neuronal subpopulations with matched RF size. (**A**) Spreads (left) and medians ± SE (right) of the RF sizes measured in each visual area (*p<0.05, **p<0.01, ***p<0.001; 1-tailed U-test, Holm-Bonferroni corrected). The number of cells in each area for which RF size could be estimated is reported on the corresponding bar of the right chart. Spreads and medians of the response latencies are reported in *Figure 7—figure supplement 1*. (**B**) For each visual area, the generalization performances achieved by single neurons (same data of *Figure 6B*) are plotted against the RF sizes (same data of panel A). In the case of LI and LL, these two metrics were significantly anti-correlated (**p<0.01; 2-tailed t-test). (**C**) Median generalization performances (± SE) achieved by single neurons, as in *Figure 6B*, but considering only neuronal subpopulations with matched RF size ranges, indicated by the gray patches in (**B**) (*p<0.05, **p<0.01, ***p<0.001; 1-tailed U-test, Holm-Bonferroni corrected). The number of neurons in each area fulfilling this constraint is reported on the corresponding bar. (**D**) Median fraction of luminance-independent stimulus information (± SE) conveyed by neuronal firing as in *Figure 3C*, but including only the RF-matched subpopulations used in (**C**). Significance levels/test as in (**C**).

The following figure supplement is available for figure 7:

**Figure supplement 1.** Response latencies.

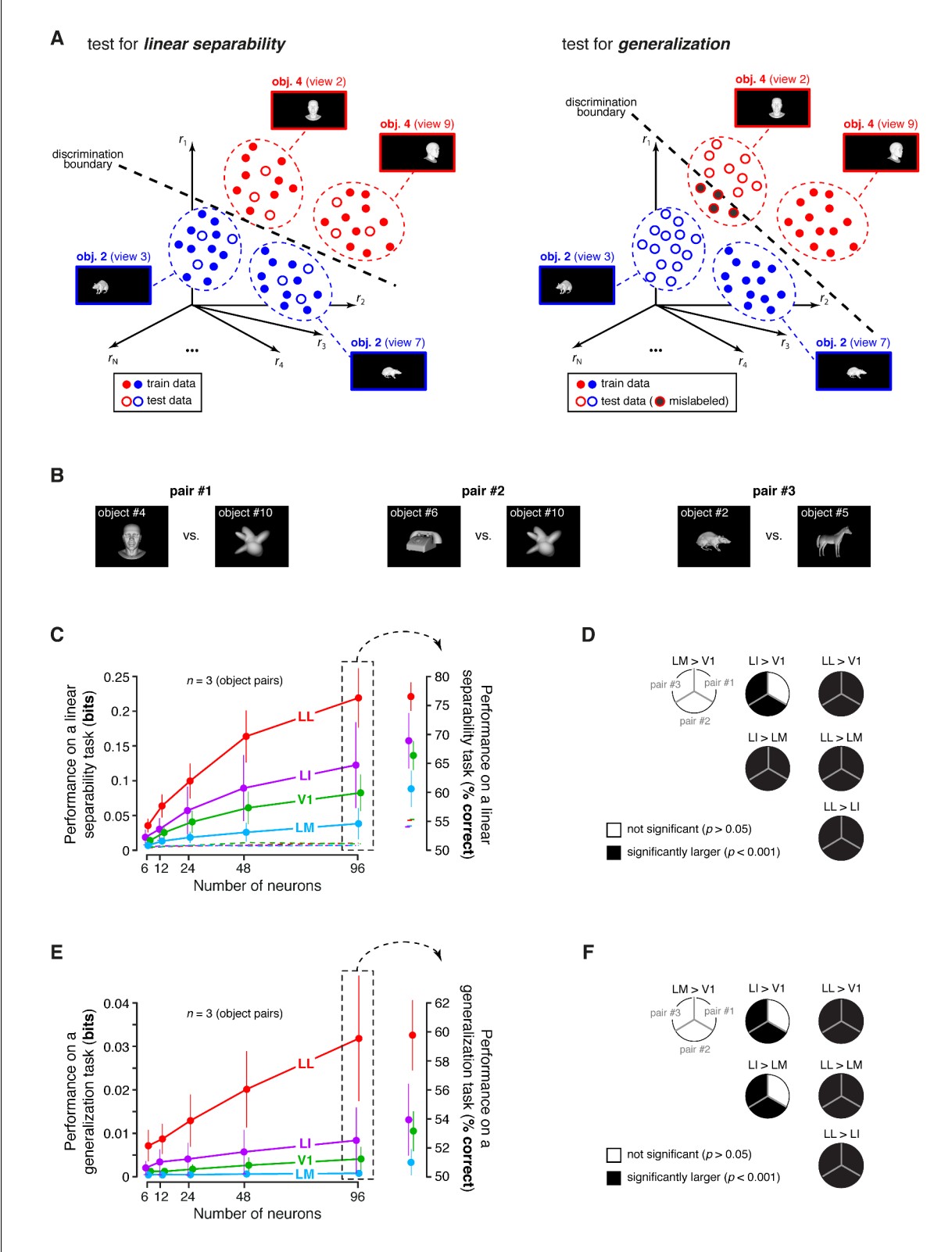

**Figure 8.** Linear separability and generalization of object representations, tested at the level of neuronal populations. (**A**) Illustration of the population decoding analyses used to test linear separability and generalization. The clouds of dots show the sets of response population vectors produced by different views of two objects. Left: for the test of linear separability, a binary linear decoder is trained with a fraction of the response vectors (filled dots) to all the views of both objects, and then tested with the left-out response vectors (empty dots), using the previously learned discrimination

*Figure 8 continued on next page*

*Figure 8 continued*

boundary (dashed line). The cartoon depicts the ideal case of two object representations that are perfectly separable. Right: for the test of generalization, a binary linear decoder is trained with all the response vectors (filled dots) produced by a single view per object, and then tested for its ability to correctly discriminate the response vectors (empty dots) produced by the other views, using the previously learned discrimination boundary (dashed line). As illustrated here, perfect linearly separability does not guarantee perfect generalization to untrained object views (see the black-filled, mislabeled response vectors in the right panel). (B) The three pairs of visual objects that were selected for the population decoding analyses shown in C-F, based on the fact that their luminance ratio fulfilled the constraint of being larger than $Th_{LumRatio} = 0.8$ for at least 96 neurons in each area. (C) Classification performance of the binary linear decoders in the test for linear separability, as a function of the number of neurons $N$ used to build the population vector space. Performances were computed for the three pairs of objects shown in (B). Each dot shows the mean of the performances obtained for the three pairs ($\pm$ SE). The performances are reported as the mutual information between the actual and the predicted object labels (left). In addition, for $N = 96$, they are also shown in terms of classification accuracy (right). The dashed lines (left) and the horizontal marks (right) show the linear separability of arbitrary groups of views of two objects (same three pairs used in the main analysis; see Results). (D) The statistical significance of each pairwise area comparison, in terms of linear separability, is reported for each individual object pair (1-tailed U-test, Holm-Bonferroni corrected). In the pie charts, a black slice indicates that the test was significant (p<0.001) for the corresponding pairs of objects and areas (e.g., LL > LI). (E) Classification performance of the binary linear decoders in the test for generalization across transformations. Same description as in (C). (F) Statistical significance of each pairwise area comparison, in terms of generalization across transformations. Same description as in (D). The same analyses, performed over a larger set of object pairs, after setting $Th_{LumRatio} = 0.6$, are shown in *Figure 8—figure supplement 1*. The dependence of linear separability and generalization from $Th_{LumRatio}$ is shown in *Figure 8—figure supplement 2*. The statistical comparison between the performances achieved by a population of 48 LL neurons and V1, LM and LI populations of 96 neurons is reported in *Figure 8—figure supplement 3*.

The following figure supplements are available for figure 8:

**Figure supplement 1.** Linear separability and generalization of object representations, tested at the level of neuronal populations, using a larger set of object pairs.

**Figure supplement 2.** Dependence of linear separability and generalization, measured at the neuronal population level, from the luminance difference of the objects to discriminate.

**Figure supplement 3.** Statistical comparison between the performance achieved by a population of 48 LL neurons and the performances yielded by populations of 96 neurons in V1, LM and LI.

the small number of object pairs that could be tested in this analysis (3), the significance of each pairwise area comparison was assessed at the level of every single pair – e.g., we tested if the objects belonging to a pair were better separable in the LL than in the LI representation (this test was performed for $N = 96$, using, as a source of variability for the performances, the 50 resampling iterations carried out for each object pair; see Materials and methods). For all object pairs, the performances yielded by LL were significantly larger than in any other area (last column of *Figure 8D*). In the case of LI, the performances were significantly larger than in LM and V1, respectively, for all and two out of three pairs (middle column of *Figure 8D*). Following the same rationale of a recent primate study (*Rust and Dicarlo, 2010*), we also checked how well binary linear classifiers could separate the representations of two arbitrary groups of object views – i.e., with each group containing half randomly-chosen views of one of the objects in the pair, and half randomly-chosen views of the other object (Materials and methods). For all the areas, the resulting discrimination performances were barely above the chance level (i.e., 0 bits and 50% correct discrimination; see dashed lines in *Figure 8C*). This means that rat lateral visual areas progressively reformat object representations, so as to make them more suitable to support *specifically* the discrimination of visual objects across view changes, and not *generically* the discrimination of arbitrary image collections.

In the test of generalization to novel object views, the classification performances (*Figure 8E*, left) were about one order of magnitude smaller than those obtained in the test of linear separability (*Figure 8C*, left). Still, for $N = 96$, they were about one order of magnitude larger than those obtained in the generalization task for single neurons (compare to *Figure 6B*). In LL, the performance increased very steeply as a function of $N$, reaching ~0.032 bits for $N = 96$, which was more than three times larger than what obtained in LI. This implies a macroscopic advantage of LL, over the other areas, in terms of generalization ability – when measured in terms of accuracy (*Figure 8E*, right), the performances in V1, LM and LI were barely above the 50% chance level, while, in LL, they approached 60% correct discrimination. This achievement is far from trivial, given how challenging

the discrimination task was, requiring generalization from a single view per object to many other views, spread across five different variation axes. Again, the statistical significance of each pairwise area comparison was assessed at the level of the individual object pairs. For all the pairs, the performances yielded by LL were significantly larger than in any other area (last column of *Figure 8F*), while, for LI, the performances were significantly larger than in LM and V1 for two out of three pairs (middle column of *Figure 8F*).

To check the generality of our conclusions, we repeated these decoding analyses after loosening the constraint on the luminance ratio, i.e., after lowering $Th_{LumRatio}$ to 0.6, which yielded a larger number of object pairs (23). Linear separability and generalization still largely followed the trends shown in *Figure 8C and E*, with LL yielding the largest performances, and LM the lowest ones (*Figure 8—figure supplement 1*). The main difference was that the performances in V1 were larger, reaching the same level of LI. This was expected, since lowering $Th_{LumRatio}$ made it easier for V1, given its sharp tuning for luminosity, to discriminate the object pairs based on their luminance difference. In fact, it should be noticed that whatever cue is available to single neurons to succeed in an invariant discrimination task, that same cue will also be effective at the population level, because the population will inherit the sensitivity to the cue of its constituent neurons. This was the case of the luminance difference between the objects in a pair, which, unless constrained to be minimal, acted as a transformation-invariant cue for the V1, LM and LI populations. This is shown by the dependence of linear separability and generalization, in these areas, on $Th_{LumRatio}$, while, for the LL population, both metrics remained virtually unchanged as a function of $Th_{LumRatio}$ (*Figure 8—figure supplement 2*). Hence, the need of matching as closely as possible the luminance of the objects, while assessing transformation tolerance, also at the population level.

Finally, we asked to what extent the decoding performances observed at the population level could be explained by the single-neuron properties illustrated in the previous sections. One possibility was that the superior tolerance achieved by the LL population resulted from the larger view-invariant information carried by the individual LL neurons, as reported in *Figure 4C* (bar plot). To test whether this was the case, we matched the LL population and the other populations in terms of the total information they carried, based on the per-neuron invariant information observed in the four areas. Specifically, since in LL the invariant information per neuron was about twice as large as in V1 and LM and about 50% larger than in LI, we compared an LL population of *N* units to V1, LM and LI populations of 2*N* units. This ensured that the latter would approximately have an overall view-invariant information that was either equal to or larger than the one of the LL population. We carried out this comparison by counting how many object pairs were still better discriminated by a population of 48 LL neurons, as compared to V1, LM and LI populations of 96 units (this is equivalent to compare the second-last red point to the last green, cyan and violet points in *Figure 8C and E*). We found that, consistently with what reported when comparing populations of equal size (*Figure 8C and E*), also in this case LL yielded significantly larger performances than the other areas in all comparisons but one (see *Figure 8—figure supplement 3*). This indicates that the larger view-invariant information per neuron observed in LL is not sufficient, by itself, to explain the extent by which the LL population surpasses the other populations in the linear discriminability and generalization tasks – the better format of the LL representation plays a key role in determining its tolerance.

## Discussion

In this study, we investigated whether a functional specialization for the processing of object information is implemented along the anatomical progression of extrastriate areas (LM, LI and LL) that, in the rat brain, run laterally to V1. Our experiments revealed that many neuronal processing properties followed a gradual, largely monotonic trend of variation along the V1-LI-LL progression. Specifically, we observed: (1) a pruning of low-level information about stimulus luminosity, essentially without loss of higher-level visual information (*Figure 3*); (2) an increase of the view-invariant object information conveyed by neuronal firing (*Figure 4*); and (3) a growth in the ability of both single neurons (*Figures 5–6*) and neuronal populations (*Figure 8*) to support discrimination of visual objects in spite of transformation in their appearance.

All these trends match very closely the key computations that are expected to take place along a feed-forward object-processing hierarchy (*DiCarlo et al., 2012*). Additionally, the tuning properties underlying the invariance attained by LI and LL are remarkably consistent with those found at the

highest stages of the monkey ventral stream (*Li et al., 2009*; *Rust and Dicarlo, 2010*). For both areas, the larger separability of object representations across view changes (*Figure 5B*), and not the shear increase of RF size (*Figure 7A*), was the key factor at the root of their superior ability to code invariantly object identity (*Figure 7C*). The same applies to the trade-off between RF size and object discriminability that was found for LI and LL (*Figure 7B*), which is reminiscent of the negative relationship between tolerance and selectivity observed in primate inferotemporal cortex (IT) and area V4 (*Zoccolan et al., 2007*; *Nandy et al., 2013*; *Sharpee et al., 2013*). Critically, these findings do not imply an exact one-to-one correspondence between areas LI and LL in the rat and areas V4 and IT in the monkey. Establishing such a parallel would require a quantitative comparison between rat and monkey visual areas in terms of (e.g.) the magnitude of the decoding performances they are able to attain, which is beyond the scope of this study. In fact, achieving such comparison would require testing all the areas in both species under the same exact experimental conditions (e.g., same combination of object identities and transformations and same presentation protocol). Instead, the strength of our conclusions rests on the fact that they are based on a qualitative agreement between the trends observed in our study across the V1-LI-LL progression and those reported in the literature for the monkey ventral stream (*DiCarlo et al., 2012*).

Because of this agreement, our results provide the strongest and most systematic functional evidence, to our knowledge, that V1, LI and LL belong to a cortical object-processing pathway, with V1 sitting at the bottom of the functional hierarchy, LL at the top, and LI acting as an intermediate processing stage. With regard to LM, our data do not support a clear assignment to the ventral stream, because this area never displayed any significant advantage, over V1, in terms of higher-order processing of visual objects. Yet, the significant increase of RF size (*Figure 7A*) and response latency (*Figure 7—figure supplement 1*) from V1 to LM is consistent with the role that mouse anatomical and functional studies have suggested for this area – one step beyond V1 in the processing hierarchy, routing both ventral and dorsal information, similar to area V2 in primates (*Wang and Burkhalter, 2007*; *Wang et al., 2012*; *Glickfeld et al., 2014*; *Juavinett and Callaway, 2015*).

As mentioned in the Introduction, these findings are in agreement with recent anatomical studies of mouse visual cortex and earlier lesion studies in the rat. At the functional level, our results strongly support the involvement of LI and LL in ventral-like computations, whose neuronal correlates have remained quite elusive to date (*Huberman and Niell, 2011*; *Niell, 2011*; *Glickfeld et al., 2014*). In fact, in the mouse, LM displays properties that are more consistent with dorsal processing, such as preference for high temporal frequencies (TFs) and low spatial frequencies (SFs) and tuning for global motion, while LI, in spite of its preference for high SFs, prefers high TFs too and only shows a marginally larger orientation selectivity than V1 (*Andermann et al., 2011*; *Marshel et al., 2011*; *Juavinett and Callaway, 2015*). Evidence of ventral-stream processing is similarly limited in the rat, despite a recent attempt at investigating LM, LI and LL, as well as the visual part of posterior temporal association cortex (named TO by the authors), with both parametric stimuli (gratings) and visual shapes (*Vermaercke et al., 2014*). Orientation and direction tuning were found to increase along the V1-LM-LI-LL-TO progression. However, when linear decoders were used to probe the ability of these areas to support shape discrimination across a position shift, only TO displayed an advantage over the other areas, and just in relative terms – i.e., the drop of discrimination performance from the train to the test position was the smallest in TO, but, in absolute terms, neither TO nor any of the lateral areas afforded a better generalization than V1 across the position change. Such weaker evidence of ventral stream processing, compared to what found in our study (e.g., see *Figure 8*), is likely attributable to the much lower number of transformations tested by (*Vermaercke et al., 2014*) (only one position change), and to the fact that sensitivity to stimulus luminance was not taken into account when comparing the areas in terms of position tolerance.

## Validity and implications of our findings

A number of control analyses were performed to verify the solidity of our conclusions. The loss of energy-related stimulus information and the increase of transformation tolerance across the V1-LI-LL progression were found: (1) across the whole cortical thickness (*Figure 3—figure supplement 1* and *Figure 6C*); (2) after matching the RF sizes of the four populations (*Figure 7C–D*); (3) across the whole spectrum of spike isolation quality in our recordings (*Figure 3—figure supplement 2A–B* and *Figure 6—figure supplement 2A–B*); and (4) when considering only neuronal subpopulations with the best spike isolation quality (*Figure 3—figure supplement 2C* and *Figure 6—figure supplement*

2C), which also equated them in terms of firing rate (*Figure 6—figure supplement 3B*). This means that no inhomogeneity in the sampling from the cortical laminae, in the amount of visual field coverage, in the quality of spike isolation, or in the magnitude of the firing rate could possibly account for our findings.

Another issue deserving a discussion is the potential impact of an inaccurate estimate of the neuronal RFs on the calculation of the RF luminance, a metric that played a key role in all our analyses. Two orders of problems emerge when estimating the RF of a neuron. First, the structure and size of the RF depend, in general, on the shape and size of the stimuli used to map it. In our experiments, we used very simple stimuli (high-contrast drifting bars) presented over a dense grid of visual field locations. The rationale was to rely on the luminance-driven component of the neuronal response to simply estimate what portion of the visual field each neuron was sensitive too. This approach was very effective, because all the recorded neurons retained some amount of sensitivity to luminance, even those that were tuned to more complex visual features than luminance alone, as the LL neurons (see the white portion of the bars in *Figure 3B*). As a result, very sharp RF maps were obtained in all the areas (see examples in *Figure 1C*). Another problem, when estimating RFs, is that they do not always have an elliptical shape, although, in many instances, they are very well approximated by 2-dimensional Gaussians (*Op De Beeck and Vogels, 2000*; *Brincat and Connor, 2004*; *Niell and Stryker, 2008*; *Rust and Dicarlo, 2010*). In our study, we took two measures to prevent poor elliptical fits from possibly affecting our conclusions. In the RF size analysis (*Figure 7*), we only included data from neurons with RFs that were well fitted by 2-dimensional Gaussians (see Materials and methods). More importantly, for the computation of the RF luminance, we did not use the fitted RFs, but we directly used the raw RF maps (see Materials and methods and *Figure 1—figure supplement 2C*). This allowed weighting the luminance of the stimulus images using the real shapes of the RFs, thus reliably computing the RF luminance for all the recorded neurons.

Finally, it is worth considering the implications of having studied object representations in anesthetized rats, passively exposed to visual stimuli. Two motivations are at the base of this choice. First, the need of probing visual neurons with the repeated presentation (tens of trials; see Materials and methods) of hundreds of different stimulus conditions, which are essential to properly investigate invariant object representations. The second motivation was the need of excluding potential effects of top-down signals, task- and state-dependence, learning and memory (*Gavornik and Bear, 2014*; *Cooke and Bear, 2015*; *Burgess et al., 2016*), which are all detrimental when the goal is to understand the initial, largely reflexive, feed-forward sweep of activation through a visual processing hierarchy (*DiCarlo et al., 2012*). For these reasons, many primate studies have investigated ventral stream functions in anesthetized monkeys [e.g., see (*Kobatake and Tanaka, 1994*; *Ito et al., 1995*; *Logothetis et al., 1999*; *Tsunoda et al., 2001*; *Sato et al., 2013*; *Chen et al., 2015*)] or, if awake animals were used, under passive viewing conditions [e.g., see (*Pasupathy and Connor, 2002*; *Brincat and Connor, 2004*; *Hung et al., 2005*; *Kiani et al., 2007*; *Willmore et al., 2010*; *Rust and Dicarlo, 2010*; *Hong et al., 2016*; *El-Shamayleh and Pasupathy, 2016*)].

In the face of these advantages, anesthesia has several drawbacks. It can depress cortical activity, especially in high-order areas (*Heinke and Schwarzbauer, 2001*), and put the cortex in a highly synchronized state (*Steriade et al., 1993*). In our study, we took inspiration from previous work on the visual cortex of anesthetized rodents (*Zhu and Yao, 2013*; *Froudarakis et al., 2014*; *Pecka et al., 2014*), cats (*Busse et al., 2009*) and monkeys (*Logothetis et al., 1999*; *Sato et al., 2013*), and we limited the impact of these issues by combining a light anesthetic with fentanyl-based sedation (Materials and methods). This yielded robust visually-evoked responses both in V1 and extrastriate areas (see PSTHs in *Figure 2A–B*, top). Still, we observed a gradual reduction of firing rate intensity along the areas' progression (*Figure 6—figure supplement 3A*), but such decrease, as mentioned above, did not account for our findings (see *Figure 6—figure supplement 3B*, *Figure 3—figure supplement 2C* and *Figure 6—figure supplement 2C*). Obviously, this does exclude the impact of anesthesia on more subtle aspects of neuronal processing. For instance, isoflurane and urethane anesthesia have been reported to alter the excitation-inhibition balance that is typical of wakefulness, thus resulting in very time persistent stimulus-evoked responses, broad RFs and reduced strength of surround suppression (*Haider et al., 2013*; *Vaiceliunaite et al., 2013*). However, under fentanyl anesthesia, surround suppression was found to be very robust in mouse V1 (*Pecka et al., 2014*), and our own recordings show that responses and RFs were far from sluggish and very similar to those obtained, from the same cortical areas, in awake rats (*Vermaercke et al., 2014*) – sharp

tuning was observed in both the time and space domains, with transient responses, rarely lasting longer than 150 ms (see examples in *Figure 2A–B*, top), and well-defined RFs (see examples in *Figure 1C*), some as small as 5° of visual angle (*Figure 7A*). Finally, neuronal activity in mouse visual cortex during active wakefulness has been shown to be very similar to that in the anesthetized state, with regard to a number of key tuning properties. These include the sharpness of orientation tuning in V1 (*Niell and Stryker, 2008*, *2010*), the sparseness and discriminability of natural scene representations in V1, LM and anterolateral area AL (*Froudarakis et al., 2014*), and the integration of global motion signals in rostrolateral area RL (*Juavinett and Callaway, 2015*).

Taken together, the evidence reviewed above is highly reassuring with regard to the validity and generality of our findings. Obviously, being our data collected in passively viewing rats, the increase of transformation tolerance observed along the V1-LI-LL progression is not the result of a supervised learning process. Rather, our findings suggest that lateral extrastriate areas act as banks of general-purpose feature detectors, each endowed with an intrinsic degree of transformation tolerance. By virtue of their larger invariance, the detectors at the highest stages are able to automatically support transformation-tolerant recognition, without the need of explicitly learning the associative relations among all the views of an object. These conclusions are in full agreement with a recent behavioral study, showing that rats are capable of spontaneously generalize their recognition to previously unseen views of an object, without the need of any training (*Tafazoli et al., 2012*).

Another important implication of our study concerns the increase of the invariant object information *per neuron* found across rat lateral visual areas (*Figure 4C*, bar plot). Critically, this finding does not imply that the overall invariant information *per area* also increases from V1 to LL. In fact, if the areas' progression acted as a purely feed-forward processing chain, a *per area* increase would mean that new object information is created from one processing step to the next, a fact that would violate the data processing inequality (*Cover and Thomas, 2006*). But in rat visual cortex, there is a strong reduction of the number of neurons from V1 through the succession of lateral extrastriate areas. LM, LI and LL occupy a cortical surface that is, respectively, 31%, 3.5% and 2.1% of the surface of V1 (*Espinoza and Thomas, 1983*). Therefore, the object information *per neuron* can increase along the areas' progression, without this implying that the total object information *per area* also increases. In addition, the connectivity among rat lateral visual areas is far from being strictly feed-forward. In both rats (*Sanderson et al., 1991*; *Montero, 1993*; *Coogan and Burkhalter, 1993*) and mice (*Wang et al., 2012*), many corticortical and thalamocortical 'bypass' routes reach higher-level visual areas, in addition to the main anatomical route that connects consecutive processing stages step-by-step (e.g., V1 directly projects to LI and LL, and LL receives direct projections from the thalamus). Thus, the concentration of more object information in each individual neuron, while the visual representation is reformatted across consecutive processing stages, should not be interpreted as an indication that total object information increases across areas. Rather, our interpretation is that the increase of invariant object information per neuron is likely an essential step to make information about object identity gradually more explicit, and more easily readable by downstream neurons that only have access to a limited number of presynaptic units (as confirmed by the linear decoding analyses shown in *Figures 5–8*).

To conclude, we believe that these results pave the way for the exploration of the neuronal mechanisms underlying invariant object representations in an animal model that is amenable to a large variety of experimental approaches (*Zoccolan, 2015*). In addition, the remarkable similarity between the anatomical organization of rat and mouse visual cortex suggests that mice too can serve as powerful models to dissect ventral stream computations, given the battery of genetic and molecular tools that this species affords (*Luo et al., 2008*; *Huberman and Niell, 2011*; *Katzner and Weigelt, 2013*).

## Materials and methods

### Animal preparation and surgery

All animal procedures were in agreement with international and institutional standards for the care and use of animals in research and were approved by the Italian Ministry of Health: project N. DGSAF 22791-A (submitted on Sep. 7, 2015) was approved on Dec. 10, 2015 (approval N. 1254/2015-PR); project N. 5388-III/14 (submitted on Aug. 23, 2012) and project N. 3612-III/12 (submitted

on Sep. 15, 2009) were approved according to the legislative decree 116/92, article 7. We used 26 naïve Long-Evans male rats (Charles River Laboratories), with age 3–12 months and weight 300–600 grams. The rats were anesthetized with an intraperitoneal (IP) injection of a solution of fentanyl (Fentanest: 0,3 mg/kg; Pfizer) and medetomidin (Domitor: 0,3 mg/kg; Orion Pharma). Body temperature was maintained at 37.5°C by a feedback-controlled heating pad (Panlab, Harvard Apparatus). Heart rate and oxygen level were monitored through a pulse oximeter (Pulsesense-VET, Nonin), and a constant flow of oxygen was delivered to the rat to prevent hypoxia.

The anesthetized animal was placed in a stereotaxic apparatus (Narishige, SR-5R). Following a scalp incision, a craniotomy was performed over the left hemisphere (~1.5 mm wide in diameter) and the dura was removed to allow the insertion of the electrode array. Stereotaxic coordinates for V1 recordings ranged from −5.16 to −7.56 mm anteroposterior (AP), with reference to bregma; for extrastriate areas (LM, LI and LL), they ranged from −6.42 to −7.68 mm AP. The exposed brain surface was covered with saline to prevent drying. The eyes were protected from direct light and prevented from drying by application of the ophthalmic solution Epigel (Ceva Vetem).

Once the surgical procedure was completed, the rat was maintained in the anesthetized state by continuous IP infusion of the fentanyl/medetomidin solution (0,1 mg/kg/h). The level of anesthesia was periodically monitored by checking the absence of tail, ear and paw reflex. The right eye of the animal was immobilized using an eye-ring anchored to the stereotaxic apparatus (the left eye was covered with black tape), with the pupil's orientation set at 0° elevation and 65° azimuth. The stereotax was positioned, so as to align the eye with the center of the stimulus display, and was rotated leftward of 45°, so as to bring the binocular field of the right eye to cover the left side of the display.

## Neuronal recordings

Recordings were performed with different configurations of 32-channel silicon probes (NeuroNexus Technologies). To maximize the coverage of the monitor by V1 RFs, neurons in this area were recorded using 8-shank arrays with either 177 $\mu m^2$ site area and 100 $\mu m$ spacing (model A8 × 4-2mm100-200-177) or 413 $\mu m^2$ site area and 50 $\mu m$ spacing (model A8 × 4-2mm50-200-413), and 4-shank arrays with 177 $\mu m^2$ site area and 100 $\mu m$ spacing (model A4 × 8–5 mm-100-200-177). To map the retinotopy along extrastriate areas (*Figure 1C*), recordings from LM, LI and LL were performed using single-shank probes with either 177 $\mu m^2$ site area and 25 $\mu m$ spacing (model A1 × 32-5mm25-177) or 413 $\mu m^2$ site area and 50 $\mu m$ spacing (model A1 × 32-5mm50-413). For V1, the probe was inserted perpendicularly to the cortex (*Figure 1—figure supplement 1C*), while, for the lateral areas, it was tilted with an angle of ~30° (*Figure 1A* and *Figure 1—figure supplement 1A–B*). For the probes with 25 $\mu m$ spacing, only half of the channels (i.e., either odd or even) were used, to avoid considering as different units the same neuron recorded by adjacent sites. To allow the histological reconstruction of the electrode insertion track, the electrode was coated, before insertion, with Vybrant DiI cell-labeling solution (Life Technologies), and, at the end of the recording session, an electrolytic lesion was performed, by passing a 5 $\mu A$ current for 2 s through the last 2 (multi-shank probes) or 4 (single-shank probe) channels at the tip of each shank (see below for a detailed description of the histological procedures).

To decide how many neurons to record in each area, we took inspiration from previous population coding studies that have compared different ventral stream areas in terms of their ability to support object recognition (*Rust and Dicarlo, 2010*; *Pagan et al., 2013*). These studies show that pairwise area comparisons become macroscopic when the size of the neuronal population in each area approaches 100 units. Therefore, in our experiments, we aimed at recording more than 100 units for each of the four visual areas under investigation. The final number of units obtained per area depended on the yield of each individual recording session (reported in *Figure 1—source data 1*) and on how accessible to recording any given area was – e.g., recordings from the deepest area (LL) were the most challenging and, since LI and LL were typically recorded simultaneously (see *Figure 1—source data 1*), we collected a large number of units from LI in the attempt of adequately sampling LL. The final number of units recorded in each area ranged from 131 (LM) to 260 (LI) (note that these numbers refer to the visually driven and stimulus informative units; see below for an explanation)

Extracellular signals were acquired using a system three workstation (Tucker-Davis Technologies) with a sampling rate of 25 kHz and were filtered from 0.3 to 5 kHz. Action potentials (spikes) were detected and sorted for each recording site separately, using Wave Clus (*Quiroga et al., 2004*) in

MATLAB (The MathWorks). Spike isolation quality was assessed using two apposite metrics (see below). Neuronal responses were quantified by counting spikes in neuron-specific spike-count windows (e.g., see gray patches in *Figure 2A–B*), with a fixed duration of 150 ms and an onset that was equal to the latency of the neuronal response, so as to capture most of the stimulus-evoked activity. The latency was estimated as the time, relative to the stimulus onset, when the response reached 20% of its peak value [for a full description see (*Zoccolan et al., 2007*)].

## Histology

At the end of the recording session, each animal was deeply anesthetized with an overdose of urethane (1.5 gr/kg) and perfused transcardially with phosphate buffer saline (PBS) 0.1 M, followed by 4% paraformaldehyde (PFA) in PBS 0.1 M, pH 7.4. The brain was extracted from the skull and postfixed overnight in 4% PFA at 4°C. After postfixation, the tissue was cryoprotected by immersion in 15% w/v sucrose in PBS 0.1 M for at least 24 hr at 4°C, and then kept in 30% w/v sucrose in PBS 0.1 M, until it was sectioned coronally at either 20 or 30 μm thickness on a freezing microtome (Leica SM2000R, Nussloch, Germany). Sections were mounted immediately on Superfrost Plus slides and let dry at room temperature overnight. A brief wash in distilled water was performed, to remove the excess of crystal salt sedimented on the slices, before inspecting them at the microscope.

For each slice, we acquired three different kinds of images, using a digital camera adapted to a Leica microscope (Leica DM6000B-CTR6000, Nussloch, Germany). First, various bright field photographs were taken at 2.5X and 10X magnification, so as to fully tile the region of the slice (left hemisphere) where the visual cortical areas were located. Second, for each of such bright field pictures, a matching fluorescence image was also acquired (using a red filter with emission at 700 nm), to visualize the red fluorescence track left by the insertion of the probe (that had been coated with Vybrant DiI cell-labeling solution before the insertion) (*DiCarlo et al., 1996*; *Blanche et al., 2005*). Following the acquisition of this set of images, the slices were further stained for Nissl substance (using the Cresyl Violet method) and pictures were taken at 2.5X and 10X magnification. In addition, to better visualize the anatomic structures within the slice, a lower magnification picture of the entire left hemisphere was also taken, using a Canon 6D digital camera with a Tamron 90 mm f/2.8 macro lens.

The fluorescence, bright field, and Nissl-stained images were processed using Inkscape (an open source SVG editor; http://www.inkscape.org), so as to reconstruct the anteroposterior coordinate of the probe insertion track and, when possible, the laminar location of the recording sites along the probe. This was achieved in three steps. First, each pair of matching fluorescence and bright field images were superimposed to produce a single picture, showing both the insertion track of the probe and the anatomical structure of the section. Then, these pictures and the Nissl-stained images were aligned by matching anatomical landmarks (e.g., the margins of the section). Finally, for each section, the mosaic of matched fluorescence, bright field and Nissl images were stitched together (by relying, again, on anatomical landmark), and then superimposed to the low-magnification image taken with the Canon digital camera. In the resulting image (see examples in *Figure 1—figure supplement 1*), the position of the shank(s) of the probe was drawn (thick black lines in *Figure 1—figure supplement 1*), by tracing the outline of the fluorescent track, and taking into account, when available, the location of the electrolytic lesion performed at the end of the recording session. Based on the known geometry of the silicon probes, the location of each recording site was also drawn over the shank(s) (yellow dots over the black lines in *Figure 1—figure supplement 1*). In the final image, the boundaries between the cortical layers (red lines in *Figure 1—figure supplement 1*) were identified, based on the difference in size, morphology and density of the Nissl-labeled cells across the cortical thickness. This allowed estimating the laminar location of the recording sites. Since the number of neurons sampled from each individual cortical layer was not very large, in our analyses, we grouped the units recorded from layers II/III and IV and those recorded from layers V and VI (see *Figure 6C* and *Figure 3—figure supplement 1*).

In some sessions, multiple recordings blocks were performed at different depths along the same probe insertion. In these cases, it was not always possible to histologically reconstruct the probe location within each block, given that a single fluorescent track was often observed, without clean-cut interruptions that could serve as landmarks for the individual blocks. In all such cases, the laminar position of the recorded units was not assigned. This explains why, in each area, the sum of the layer-labeled cells reported in our analyses is not equal to the total number of recorded units (e.g., compare the numbers of neurons reported in *Figure 6B* to those reported in *Figure 6C*). Since, in

multi-block recording sessions, the probe reconstruction was especially difficult for the initial (more superficial) block, LM (the first area to be recorded during oblique penetrations) was the area in which, for a relative large fraction of neurons, we did not assign a laminar position.

## Visual stimuli

Our stimulus set consisted of 380 visual object conditions, obtained by producing 38 distinct views (or transformations) of 10 different objects, using the ray tracer POV-Ray (http://www.povray.org/). The objects were chosen, so as to span a range of shape features and low-level properties (e.g., luminance). Six of them (#1–6) were computer-graphics reproductions of real-world objects, while the remaining four (#7–10) were artificial shapes, originally designed for a behavioral study assessing invariant visual object recognition in rats (*Tafazoli et al., 2012*). *Figure 1B* (top) shows these objects, as they appeared in the *pose* that we defined as *default*, with regard to the rotation parameters (i. e., 0° in-plane and in-depth rotation), and at their *default luminosity* (i.e., 100% luminance). Other default parameters were the *default size* (35° of visual angle) and *two default azimuth positions* (−15° and +15° of visual angle) over the stimulus display. Each of these default parameters was held constant, when some other parameter was changed to generate the object transformations, as detailed below. The elevation of the stimuli was fixed at 0° of visual angle and never varied. Size was computed as the diameter of an object's bounding circle. The transformations that each object underwent were the following.

*Positions changes:* each object was horizontally shifted across the stimulus display, from −22.5° to +30° of visual angle (azimuth), in steps of 7.5° (*Figure 1B.1*). These numbers refer to the center of the display (0°), which was aligned to the position of the right eye of the rat (i.e., a hypothetical straight line passing through the eye and perpendicular to the monitor would hit it exactly in the middle). The pose, luminance and size of the objects were the default ones.

*Size changes:* each object was scaled from 15° to 55° of visual angle, in steps of 10° (*Figure 1B.2*). The pose, luminance and positions (−15° and +15°) of the objects were the default ones.

*In-plane rotations:* each object was in-plane rotated from −40° to +40°, in steps of 20° (*Figure 1B.3*). The in-depth rotation, luminance, size and positions of the objects were the default ones.

*In-depth rotations:* each object was presented at five different in-depth (azimuth) rotation angles: −60°, −40°, 0°, + 40°, and +60° (*Figure 1B.4*). The in-plane rotation, luminance, size and positions of the objects were the default ones.

*Luminance changes:* each object was presented at four different luminance levels: 100% (i.e., default luminance), 50%, 25% and 12.5% (*Figure 1B.5*). The pose, size and positions of the objects were the default ones.

The object conditions were presented in rapid sequence (250 ms stimulus on, followed by 250 ms of blank screen), randomly interleaved with the drifting bars used to map the neuronal RFs (*Figure 1—figure supplement 2A–B*). During a recording session, the presentation of each condition was repeated a large number of times. For V1 neurons, each conditions was presented, on average, 26 ± 2 times; for LM neurons, 25 ± 4 times; for LI neurons, 27 ± 3 times; and for LL, neurons 28 ± 2 times. This allowed obtaining a good estimate of the conditional probability $P(R|O \& T)$ of measuring a given response $R$ to any combination of object identity $O$ and transformation $T$.

The stimuli were displayed on a 47 inch LCD monitor (Sharp PN-E471R), with 1920 × 1080 pixel resolution, 60 Hz refresh rate, 9 ms response time, 700 cd/m2 maximum brightness, 1.200:1 contrast ratio, positioned at a distance of 30 cm from the right eye, spanning a visual field of 120° azimuth and 90° elevation. To avoid distortions of the objects' shape, the stimuli were presented under a tangent screen projection (see next section and *Figure 1—figure supplement 2D–F*).

## Tangent screen projection

As explained above, the stimulus display was positioned at a distance of 30 cm from the rat eye. This number was chosen, because the optimal viewing distance for rats has been reported to range between 20 and 30 cm (*Wiesenfeld and Branchek, 1976*). In addition, earlier behavioral studies from our group have shown that rats are capable of discriminating complex visual objects under a variety of identity-preserving transformations, when viewing the objects at a distance of 30 cm (*Zoccolan et al., 2009*; *Tafazoli et al., 2012*; *Alemi-Neissi et al., 2013*; *Rosselli et al., 2015*).

However, because of such a short viewing distance, objects will appear distorted, when they undergo large translations over the stimulus display. Because of this, position changes will also result in size changes and distortions of the objects' shape, unless appropriate corrections are applied. To address this issue, we displayed the stimuli under a tangent screen projection. This projection allows presenting the stimuli as they would appear, if they were shown on virtual screens that are tangent to a circle centered on the rat's eye, with a radius equal to the distance from the eye to the point on the display just in front of it (i.e., 30 cm). Thanks to this projection, the shape, size and aspect ratio of each stimulus were preserved across all the eight azimuth positions tested in our experiment (see previous section).

The tangent projection is explained in *Figure 1—figure supplement 2D–F*. Panels D and E show, respectively, a top view and a side view of the rat eye and the stimulus display ($O_1$). $R$ is the distance between the eye and the center of the display. $O_2$ is an example (virtual) tangent screen, where an object would be shown, if its center was translated of an azimuth angle $\theta$ to the left of the center of the stimulus display (while maintaining the default elevation of 0°). The coordinate $x_0$ indicates the projection of the object's center over $O_1$, following this azimuth shift $\theta$. The Cartesian coordinates ($x_2$, $y_2$) indicate the position of a pixel of the stimulus image, relative to the object's center, over the virtual screen $O_2$, while ($x_1$, $y_1$) indicate the projection of this point over the display $O_1$, i.e., its coordinates, relative to the center of the object ($x_0$, 0) in $O_1$. The red line drawn over $O_2$ shows the distance of this pixel from the object's center over the tangent screen, while the red line drawn over $O_1$ is the projection of this distance over the stimulus display.

The projection of any point ($x_2$, $y_2$) in the virtual tangent screen $O_2$ to a point ($x_1$, $y_1$) in the stimulus display $O_1$ can be computed using simple trigonometric relationships. *Figure 1—figure supplement 2D* shows how $x_1$ can be expressed as a function of $x_2$ and $\theta$:

$$x_1 = R \, \tan(\vartheta + \varphi) - x_0,$$

where:

$$x_0 = R \, \tan(\vartheta) \quad \text{and} \quad \varphi = \tan^{-1}(x_2/R).$$

Note that $\varphi$ is the azimuth position of the point ($x_2$, $y_2$), when expressed in spherical coordinates. Similarly, *Figure 1—figure supplement 2E* shows how $y_1$ can be expressed as a function of $y_2$ and $\theta$:

$$y_1 = y_2 \frac{R_1}{R_2}$$

where:

$$R_2 = \frac{R}{\cos \varphi} \quad \text{and} \quad R_1 = \frac{R}{\cos(\varphi + \vartheta)} \, .$$

To better illustrate the effect of the tangent screen projection, *Figure 1—figure supplement 2D* shows how two points at the same distance from (but on opposite sides of) the object's center in the tangent screen (see, respectively, the red and green lines over $O_2$) would be projected over the stimulus display (see, respectively, the red and green lines over $O_1$). As shown by the drawing, these two points would not be equidistant any longer from the object's center, in the projection over $O_1$. This can be better appreciated by looking, in *Figure 1—figure supplement 2F* (top), at the images of an example object (object #8) shown at positions −22.5°, −15°, −7.5° and 0° of visual angle (azimuth) over the stimulus display (these are a subset of the eight different positions tested for each object in our experiment; see above). The distortion applied to the object by the tangent screen projection becomes progressively larger and more asymmetrical (with respect to the vertical axis of the object), the larger is the distance of the object's center from the center of the display (0° azimuth). Critically, this distortion was designed to exactly compensate the one produced by the perspective projection to the retina, so that, regardless of the magnitude of the azimuth displacement, the resulting projection of an object over the retina will have the same shape, size and aspect ratio as the ones produced by the object shown at center of the display, right in front of the rat's eye (*Figure 1—figure supplement 2F*, bottom).

## Selection of the neuronal populations included in the analyses

Throughout our study, data analysis was restricted to those neurons that met two criteria: (1) being visually driven; and (2) being stimulus informative. The first criterion was implemented as following. Given a neuron, its response to each of the 380 object conditions and its background firing rate were computed. A 2-tailed t-test was then applied to check if at least one of these conditions evoked a response that was significantly different (either larger or lower) than background (p<0.05 with Bonferroni correction for multiple comparison). The second criterion was based on the computation of the mutual information between stimulus and response $I(R;S)$ (defined below). Note that, in this case, only 230 object conditions $S$ were used (see next section). To be included in the analysis, a neuron had to carry a significant amount of information $I(R;S)$ (p<0.05; permutation test; described below). The number of neurons that met these two criteria from each area per rat is reported in *Figure 1—source data 1*.

## Selection of the object conditions to be included in the mutual information and single-cell decoding analyses

As explained above, 38 different transformations of each object were presented during the experiment. These included eight positions across the horizontal extent (azimuth) of the stimulus display, plus four sizes, four in-plane rotations, 4-in-depth rotations and three luminance levels, presented at two of the eight positions (i.e., −15° and +15° azimuth). For any given neuron, all the eight positions were included in the single-cell mutual information and decoding analyses (*Figures 3–7*), while, of the remaining transformations, only those shown at the position that was closer to the neuron's RF center (i.e., either −15° or +15°) were used. This yielded 23 different transformations per object, for a total of 230 stimuli $S$. The motivation of choosing this subset of transformations was to maximize the coverage of the object conditions by the RFs of the recorded neurons. In the case of the population decoding analyses (*Figure 8*), 19 of these 23 transformations per object were used (see below).

## Computation of the receptive field size

Given a neuron, we obtained an estimate of its receptive field (RF) profile, using a procedure adapted from *Niell and Stryker (2008)*. We measured the neuronal response to 10° long drifting bars with four different orientations (0°, 45°, 90°, and 135°), presented over a grid of 6 × 11 cells on the stimulus display (*Figure 1—figure supplement 2A*). The responses to the four orientations in each cell were averaged to obtain a two-dimensional map, showing the firing intensity at each location over the display (*Figure 1—figure supplement 2B*, left). This raw RF map was fitted with a two-dimensional Gaussian, with independent widths (SDs) $\sigma_x$ and $\sigma_y$ (*Figure 1—figure supplement 2B*, right). The RF size (diameter) was computed as the average of $\sigma_x$ and $\sigma_y$ (see the black ellipse in *Figure 1—figure supplement 2B*, left). The goodness of the fit was measured by the *coefficient of determination* $R^2$, defined as:

$$R^2 = 1 - \frac{SS_{\mathrm{resid}}}{SS_{\mathrm{total}}}$$

where $SS_{\mathrm{resid}}$ is the sum of the squared residuals of the fit and $SS_{\mathrm{total}}$ is the sum of the squared differences from the mean of the raw RF values. A fit was considered acceptable if $R^2$ was larger than 0.5.

## Computation of the receptive field (RF) luminance and RF contrast

To estimate how much luminous energy any given stimulus (i.e., object condition) impinged on a neuronal RF, we defined a metric (which we called *RF luminance*), resulting from computing the dot product between the raw RF map of the neuron (normalized to its maximal value, so as to range between 0 and 1) and the luminance profile of the stimulus over the image plane. This is equivalent to weight the stimulus image by the RF profile of the neuron (*Figure 1—figure supplement 2C* provides a graphical description of this procedure). This quantity was then normalized by the maximal luminance that could possibly fall into the neuron's RF (corresponding to the case of a full-field stimulus at maximal brightness), hence the percentage values reported in *Figure 2A–B* (white font) and *Figure 2—figure supplement 1*.

A similar approach was used to measure the variability of the pattern of luminance impinged by a stimulus on a neuronal RF, using a metric that we called *RF contrast*. This metric was defined as the

standard deviation of the RF-weighted luminance values produced by the stimulus that were contained within the RF itself (see the rightmost plot in *Figure 1—figure supplement 2C*). Note that, in general, since the neuronal RFs did not fully cover all the object conditions (see examples in *Figure 2A–B*), also the background contributed to the RF contrast metric. Therefore, in the analysis in which we assessed the amount of information carried by single neurons about RF contrast (*Figure 3—figure supplement 4*), we only considered object conditions that covered at least 10% of the RFs. This restriction was applied because we wanted to measure how neurons coded the contrast over a large enough surface of the visual objects. In addition, since the number of stimuli that met this criterion was different for the various neurons, we included in the analysis only neurons for which at least 23 object conditions fulfilled this constraint (this explains why the number of neurons reported in *Figure 3—figure supplement 4A* is lower than the total number of units recorded in each area; e.g., compare to *Figure 3B*). This insured that the information about RF contrast was estimated with enough visual stimuli.

## Quality metrics of spike isolation

The quality of spike isolation was assessed through two widely used benchmarks: signal-to-noise ratio (*SNR*) and refractory violations (*RV*) (*Quiroga et al., 2005, 2008*; *Gelbard-Sagiv et al., 2008*; *Hill et al., 2011*). These metrics were defined as following.

$$SNR = \frac{A_{\mathrm{signal}}}{A_{\mathrm{noise}}},$$

where $A_{\mathrm{signal}}$ is the average peak-to-peak amplitude of all the spikes detected as a single cluster by the sorting algorithm, and $A_{\mathrm{noise}}$ is an estimate of the variability of the background noise computed from the rest of the filtered signal (in our application, this was the median of the absolute value of the filtered signal, divided by 0.6745 [*Quiroga et al., 2004*]). *RV* was defined as the fraction of inter spike intervals (ISI) that were shorter than 2 *ms*, the rationale being that a large *RV* indicates a substantial violation of the neuron's refractory period and, therefore, a contamination from other units (*Hill et al., 2011*).

These metrics were used in two ways. First, the neurons recorded from the four areas were pooled together and the resulting distributions of *SNR* and *RV* values were divided in three equi-populated bins (tertiles). Within each tertile, the mutual information and generalization metrics were re-computed, so as to compare neuronal subpopulations within the same range of isolation quality (*Figure 3—figure supplement 2A–B* and *Figure 6—figure supplement 2A–B*). Second, we selected from the four areas only neurons with good isolation quality, as defined by imposing both *SNR* >10 and *RV* <2%, and, again, we restricted the computation of the mutual information and generalization metrics to these subpopulations (*Figure 3—figure supplement 2C* and *Figure 6—figure supplement 2C*). These constraints also equated the neuronal subpopulations in terms of firing rate (*Figure 6—figure supplement 3B*), which, otherwise, would significantly decrease from V1 to the more lateral areas (*Figure 6—figure supplement 3A*).

## Statistical tests

All the single-cell properties measured for the visual areas have been reported in terms of medians ± Standard Error (SE) of the median (estimated by bootstrap). In all these cases, the statistical significance of each pairwise, between-area comparison was assessed with a 1-tailed, Mann-Whitney U-test, with Holm-Bonferroni correction for multiple comparisons. The choice of the 1-tailed test was motivated by the fact that, in any comparison, we had a clear hypothesis about the rank of each visual area, with respect to the measured property, based on previous anatomical, lesion and functional studies (see Introduction and Discussion).

## Mutual information analysis

The mutual information $I(R;S)$ between stimulus $S$ and response $R$ was computed according to *eq. 1* (see Results), and its meaning is graphically illustrated in *Figure 3A*. Here, we report some more technical details about how the calculation of the information metrics used through the study (i.e., see *Equation 1, 2 and 3*) was carried out.

In all the information theoretic analyses, the response $R$ was quantified as the number of spikes fired by a neuron in a 150 ms-wide spike count window (see above), discretized into three equi-populated bins (whose boundaries were computed, for each neuron independently, on the whole set of responses collected across repeated presentations of all the stimuli). Given that we recorded an average of 26.5 trials per stimulus (see above), this discretization yielded about nine trials per stimulus per response bin. Earlier studies (*Panzeri and Treves, 1996*; *Panzeri et al., 2007*) showed that the limited sampling bias in information calculations can be effectively corrected out – leading to precise information estimations – if there are at least four repetitions per stimulus and per response bin. Quantizing responses in three bins was thus appropriate to obtain highly conservative and unbiased information estimates. Yet, we explored a range of different bin numbers within the extent for which the bias could reliably be corrected, and we found similar patterns of mutual information values across the four visual areas.

As shown in the Results (see *Equation 2* and *Figure 3B*), $I(R; S)$ can be rewritten, using conditional mutual information (*Ince et al., 2012*), as the sum of the luminance information $I(R; L)$ and the higher-level, luminance-independent information $I(R; S'|L)$. Since, for each object and neuron, the maximum number of possible luminance values $L$ is equal to the number of unique transformations the objet underwent (i.e., 23), $L$ was discretized in 23 bins (see *Figure 2C–D*). This number of luminance bins fully covered the luminance variation for each object, without loss of luminance information, and it yielded, on average, 100 trials per response and stimulus bin, which was comfortably sufficient to control for the sampling bias in the computation of the mutual information (*Panzeri et al., 2007*). The same arguments apply to the analysis in which $I(R; S)$ was decomposed as the sum of the contrast information $I(R; C)$ and the contrast-independent information $I(R; S'|C)$ (see *Figure 3—figure supplement 4*).

To evaluate the statistical significance of mutual information values we used a random permutation method (*Ince et al., 2012*). In this method, separately for each considered information calculation and for each neuron, the association between stimulus and response was randomly permuted 100 times, to generate a null-hypothesis distribution of information values in the absence of any association between stimulus and response. The significance levels (*p* values) of the mutual information values were obtained from this null-hypothesis distribution, as explained in (*Ince et al., 2012*).

All information measures were computed using the Information Breakdown Toolbox (*Magri et al., 2009*) and were corrected for limited sampling bias using the Panzeri-Treves method (*Panzeri and Treves, 1996*; *Panzeri et al., 2007*). This method uses an asymptotic large-N expansion (where N is the total number of trials available across all stimuli) to estimate and then subtract out the limited sampling bias from the raw information estimate. The asymptotic estimation of the bias has the following expression: $BIAS = \frac{1}{2N \ln 2} \left[ \sum_s (R_s - 1) - (R - 1) \right]$ where $R_s$ and $R$ are the number of bins with non-zero stimulus-specific and stimulus-unspecific response probabilities, respectively. These numbers of bins are estimated from the data with a Bayesian procedure, as described in *Panzeri and Treves (1996)*.

## Single cell decoding analysis

This analysis was meant to assess the ability of single neuronal responses to support the discrimination of a pair of objects in spite of changes in their appearance. Given a neuron, we defined the following variables. $R$ is the neuron's response, measured by counting the number of spikes fired in a given spike count window, following stimulus presentation (see above for the choice of the spike count window). $O$ denotes the identity of a visual object, among the set of 10 available objects, i.e., $O = \{o_1, o_2, \ldots, o_{10}\}$. $T$ denotes the specific transformation an objet underwent, among the set of 23 transformations (i.e., views) available for that neuron (see above for an explanation of how these conditions were chosen for each neuron), i.e., $T = \{t_1, t_2, \ldots, t_{23}\}$. For each neuron, we selected all possible pairs of objects $o_i$ and $o_j$, and, for each pair, we performed two kinds of analysis: (1) a test of linear separability of object representations across transformations (*Figure 5*); and (2) various tests of generalization of the discrimination of the two objects across transformations (*Figure 6*). The results obtained for each object pair were then averaged, to estimate the linear separability and generalizability afforded by each neuron. Both procedures are detailed below.

## Test of linear separability

This analysis was carried out following a 5-fold cross-validation procedure. Given a neuron and a pair of objects $o_i$ and $o_j$, we randomly partitioned the set of responses of the neuron to the repeated presentation of each object (across all 23 tested views) in five subsets. In each cross-validation loop, only four of these subsets were used as the training data to build the decoder, while the remaining subset was used as the test data to measure the decoder's performance at discriminating the two objects. More formally, we considered the conditional probabilities of observing a response $R$ to the presentation of any view of each object, i.e., $P_{\mathrm{train}}(R = r | O = o_i \ \& \ T = \{t_1, t_2, \ldots, t_{23}\})$ and $P_{\mathrm{train}}(R = r | O = o_j \ \& \ T = \{t_1, t_2, \ldots, t_{23}\})$, where the 'train' subscript indicates that only 4/5 of the responses (those belonging to the training data set) were used to compute the conditional probabilities. We then used these probabilities to find a boundary, along the spike count axis, that allowed the discrimination of all the views of an object from all the views of the other object. To avoid overfitting the decoder to the training data, we minimized the assumptions on the shape of the conditional probabilities, and we simply parameterized them by computing their means $\mu_i$ and $\mu_j$ and variances $\sigma_i$ and $\sigma_j$ (with $\mu_i \geq \mu_j$ and $\sigma_i \neq \sigma_j$). We then used these parameters to find the discrimination boundary, by applying quadratic discriminant analysis (*Hastie et al., 2009*). That is, the discrimination boundary was defined as the threshold response $r_{th}$, obtained by setting to zero the log-ratio of the posterior probabilities, i.e.:

$$\log \frac{P_{\mathrm{train}}(O = o_i \ \& \ T = \{t_1, t_2, \ldots, t_{23}\} | R = r_{th})}{P_{\mathrm{train}}(O = o_j \ \& \ T = \{t_1, t_2, \ldots, t_{23}\} | R = r_{th})} = 0 \tag{4}$$

Having fixed this decision boundary, the decoder was then tested for its ability to correctly classify the remaining 1/5 of the responses belonging to the test set, according to the following binary classification rule:

$$O = \begin{cases} o_i, & \text{if } R > r_{th} \\ o_j, & \text{if } R < r_{th} \end{cases} \tag{5}$$

This procedure was repeated for all possible combinations of the five subsets of responses in four train sets and one test set, with each combination yielding a distinct cross-validation loop. The outcomes of the classification, obtained across the resulting five cross-validation loops, were collected in a confusion matrix, which was then used to compute the performance of the decoder with that specific object pair. As explained in the Results, the performance was computed as the mutual information between the actual and the predicted object labels from the decoding outcomes [i.e., as the mutual information between the rows and columns of the confusion matrix (*Quiroga and Panzeri, 2009*)]. This computation was performed for all possible object pairs. The resulting performances were then averaged to obtain the linear separability afforded by the neuron. Note that, to prevent large decoding performances from being trivially achieved only based on luminance differences, only those pairs with objects having a luminosity ratio larger than a given threshold $Th_{LumRatio}$ (see Results) were included in the final average.

## Test of generalization

Given a neuron and a pair of objects $o_i$ and $o_j$, we first trained a binary classifier to partition the spike count axis with a boundary that allowed the discrimination of two specific transformations $t_x$ and $t_y$ of the two objects. To this aim, we considered the conditional probabilities of observing a response $R$ to the presentation of the two object views, i.e., $P(R = r | O = o_i \ \& \ T = t_x)$ and $P(R = r | O = o_j \ \& \ T = t_y)$. Note that these conditional probabilities were obtained by taking into account all the responses of the neuron to the repeated presentation of that specific combination of object identity $O$ and transformation $T$. Similarly to what done in the linear separability analysis, the conditional probabilities were parameterized by computing their means $\mu_{ix}$ and $\mu_{jy}$ and variances $\sigma_{ix}$ and $\sigma_{jy}$ (with $\mu_{ix} \geq \mu_{jy}$ and $\sigma_{ix} \neq \sigma_{jy}$), and these parameters were used to find the discrimination boundary, by applying quadratic discriminant analysis (*Hastie et al., 2009*). That is, the discrimination boundary was defined as the threshold response $r_{th}$, obtained by setting to zero the log-ratio of the posterior probabilities, i.e.:

$$\log \frac{P(O = o_i \ \& \ T = t_x | R = r_{th})}{P(O = o_j \ \& \ T = t_y | R = r_{th})} = 0 \tag{6}$$

Having fixed this decision boundary, the decoder was then tested for its ability to correctly classify the responses of the neuron to different transformations (i.e., $T \neq t_x$ and $T \neq t_y$) of the same two objects $o_i$ and $o_j$, according to the following binary classification rule:

$$O = \begin{cases} o_i, & \text{if } R > r_{th} \\ o_j, & \text{if } R < r_{th} \end{cases} \tag{7}$$

Following this general scheme, the cross-validation procedure worked as following. Given a neuron, and given a pair of objects, we randomly sampled, independently for each object, one of the 23 transformations available for that neuron. The spike count distributions produced by these transformations were used to build the decision boundary $r_{th}$, as defined in *Equation (6)*. We then used the responses to the repeated presentation of the remaining 22 transformations of each object as the test set to measure the generalization performance of the decoder, according to the classification rule defined in *Equation (7)*. For each object pair, this procedure was repeated 1000 times. In each of these runs, the training transformations of the two objects were randomly sampled, so as to span a wide range of training and testing object views. This yielded an average generalization performance per object pair. This computation was performed for all possible pairs. The resulting performances were then averaged to obtain the generalization performance afforded by the neuron. As for the linear separability analysis, the pairs included in the final average were only those with objects having a luminosity ratio larger than a given threshold $Th_{LumRatio}$. Two different versions of this analysis were carried out. In the first one, the training and testing views were randomly sampled from the full set of 23 transformations available for each neuron (*Figure 6B–C*). In the second one, the training and testing views were randomly sampled from the individual variation axes: position, size, in-plane and in-depth rotations and luminance (*Figure 6D*).

## Test of generalization along parametric position and size changes

In the case of the transformations along the position axis, a second kind of analysis was also performed. This analysis was meant to assess how sharply the generalization performance of the binary decoders decayed as a function of increasingly large position changes, thus providing an estimate of the spatial extent of the invariance afforded by a neuron over the visual field (*Figure 6—figure supplement 1A*). Given a neuron, and given a pair of objects, we selected one of the eight azimuth positions tested during the experiment (*Figure 1B*, bottom). Note that here, differently from the cross-validation procedure described above, the same transformation (i.e., the same azimuth position) was selected for both objects in the pair. We used the spike count distributions produced by the two objects at this specific position to build the decision boundary $r_{th}$, as defined in *Equation 6*. We then tested the ability of this decision boundary to correctly discriminate the two objects presented at testing positions that were increasingly further apart from the training location. That is, for each testing position, we took the responses of the neuron to the repeated presentation of the two objects at that position, and we measured the generalization performance of the decoder, according to the classification rule defined in *Equation 7*. The testing positions were chosen either to the left or to the right of the selected training position, depending what direction allowed spanning at least 30° of the visual field. Specifically, for training positions between −22.5° and 0° of visual angle, four testing positions were taken to the right of the training location; vice versa, for training positions between +7.5° and +30° of visual angle, four testing positions were taken to the left of the training location. For any training position, this procedure yielded a curve, showing how stable the discriminability of the objects in the pair was across a span of 30° over the visual field. For any given object pair, this procedure was repeated across all possible training positions, starting from the leftmost one (at −22.5°), up to the rightmost one (+30°). The resulting generalization performances were averaged to yield the mean generalization curve over the position axis for that specific object pair. This computation was then repeated for all possible pairs, and the resulting performances were averaged to obtain the generalization curve over the position axis afforded by the neuron. The pairs included in this final average were those with objects having a luminosity ratio larger than 0.9 (i.e., with $Th_{LumRatio} = 0.9$). In addition, to obtain a cleaner estimate of the dependence of position

tolerance from the magnitude of the translation, the analysis was restricted to the neurons with unimodal and elliptical RF profiles, which were well fitted by 2-dimensional Gaussians (i.e., those neurons contributing to the RF statistics shown in *Figure 7A*). The resulting median generalization curves obtained for the four visual areas are shown in *Figure 6—figure supplement 1A*.

A similar kind of analysis was performed for the transformations along the size axis (see *Figure 6—figure supplement 1B*). The cross-validation procedure was similar to the one described in the previous paragraph for the position changes. However, since the size conditions were fewer than the position ones (*Figure 1B*, bottom), and lent themselves to be analyzed in terms of generalization from small to larger sizes (and vice versa), we grouped them into three classes: (1) the default size (i.e., 35°); (2) the two sizes than were smaller than 35° (i.e., 15° and 25°); and 3) the two sizes than were larger than 35° (i.e., 45° and 55°). Given an object pair, we tested all possible generalizations from the default size to the smaller and larger sizes and, vice versa. In the first case, we used the spike count distributions produced by presenting the two objects at size 35° to build the decision boundary $r_{th}$, as defined in *Equation 6*. We then tested the ability of this decision boundary to correctly discriminate the two objects presented at each of the other sizes. That is, for each testing size, we took the responses of the neuron to the repeated presentation of the two objects at that size, and we measured the generalization performance of the decoder, according to the classification rule defined in *Equation 7*. This procedure yielded four generalization performances: two from size 35° to larger sizes, and two from size 35° to smaller sizes. A similar approach was used to compute the generalization from the other sizes to the default one. Also in this case, four generalization performances were obtained: two from smaller sizes to size 35°, and two from larger sizes to size 35°. Overall, the full procedure yielded eight generalization performances per object pair. These performances were divided into two groups. The first group included the generalization from size 35° to smaller sizes and from larger sizes to size 35°. The second group included the generalization from size 35° to larger sizes and from smaller sizes to size 35°. The four performances within each group were averaged to yield the final large→small and small→large generalization performances for the object pair. This procedure was repeated for each pair and the resulting performances were averaged to estimate the generalization afforded by the neuron, when object size changed from large to small and vice versa. The pairs included in these final averages were those with objects having a luminosity ratio larger than 0.9 (i.e., with $Th_{LumRatio} = 0.9$). The resulting median generalization performances obtained for the four visual areas are shown in *Figure 6—figure supplement 1B*.

## Computation of the chance level and assessment of the statistical significance of the classification performances

In the cross-validation procedures described above (for both the tests of linear separability and generalization), each step yielding a classification performance also yielded a chance generalization performance. This was computed by randomly permuting the object identity labels associated to the responses of the neuron before building the decoder. While carrying out this permutation, care was taken to randomly shuffle only the label $O$, which specifies object identity, and not the label $T$, which specifies the transformation the objet underwent. This means that, given a pair of objects $o_i$ and $o_j$, the identity labels $o_i$ and $o_j$ were randomly shuffled among the responses produced by matching views of the two objects (i.e., the shuffling was performed, separately, for each of the 23 transformations the two objects underwent). This was done to isolate the contribution of the only variable of interest (i.e., object identity) to the performances attained by the classifiers.

For any given neuron, chance performances were obtained for all object pairs, and were then averaged, according to the same procedures described above, to yield the final chance decoding performance of the neuron. These chance performances were used in two ways. First, to obtain conservative estimates of the ability of the decoders to generalize to new object views, the chance performances were subtracted from the actual (i.e., un-shuffled) classification performances. This allowed measuring and reporting, in all the single-cell decoding analyses (*Figures 5–7*), the net amount of transformation-tolerance afforded by single neurons in the four visual areas (in general, these chance performances obtained by shuffling were all very close to the theoretical 0 bit chance level of the binary classification task). Second, the distribution of classification performances obtained for a neuronal population was compared to the matching distribution of chance performances, to assess whether the former was significantly larger than the latter (according to a 1-tailed,

Mann-Whitney U-test). This was the case for all the classification performances reported in *Figures 5–7*.

## Population decoding analysis

The population decoding analysis followed the same rationale of the single-cell decoding analysis. Binary linear decoders were built to assess the ability of a population of neurons in a visual area to support the discrimination of a pair of objects in spite of changes in their appearance. Again, two kinds of analyses were preformed – a test of linear separability of the object representations, and a test of generalization of the discrimination of the two objects across transformations.

## Construction of the population vector representational space

Following the design of previous population decoding studies of the monkey ventral stream (*Hung et al., 2005*; *Rust and Dicarlo, 2010*; *Pagan et al., 2013*), we measured whether (and how sharply) the decoders' performance grew as a function of the size of the neuronal populations, by randomly sampling subpopulations of 6, 12, 24, 48 and 96 units from the full sets of neurons recorded in each area. More specifically, for a given subpopulation size $N$, with $N = \{6, 12, 24, 48, 96\}$, we run 50 subpopulation resampling iterations. In each iteration, $N$ different units were randomly chosen from the full population, and, for each stimulus condition $S$ (i.e., each combination of object identity and transformation $S = O\&T$), $M$ pseudo-population response vectors were built, drawing from the responses of the $N$ units to the repeated presentation of $S$ (in the cartoons of *Figure 8A*, a set of pseudo-population vectors corresponding to a given stimulus $S$ is graphically illustrated as a cloud of dots with a specific color). Since the neurons belonging to each area were recorded across different sessions (and, therefore, not all the neurons in a given area were recorded simultaneously; see *Figure 1—source data 1*), the pseudo-population response vectors to stimulus $S$ were built by assigning to each component of the vector the response of the corresponding neuron in a randomly sampled presentation of $S$. For each component, $M$ of such responses were sampled without replacement, to obtain the final set of $M$ pseudo-population vectors to be used in a given subpopulation resampling iteration. In each resampling iteration, linear separability and generalization were computed as detailed below.

## Test of linear separability

As in the case of the single-cell analysis, we applied a 5-fold cross-validation procedure. Given a neuronal subpopulation of size $N$ (obtained in a specific resampling iteration) and a pair of objects $o_i$ and $o_j$, we randomly partitioned the set of pseudo-population response vectors associated to the presentation of each object (across all 23 tested views) in five subsets. In each cross-validation loop, only four of these subsets were used as the train data to build the decoder, while the remaining subset was used as the test data to measure the decoder's performance at discriminating the two objects. Based on the train data, a linear decoder learned to find a hyperplane that partitioned the population vector space into two semi-spaces, each corresponding to a specific object label (i.e., either $o_i$ or $o_j$).

As a decoder, we used a Support Vector Machine (SVM) (*Cristianini and Shawe-Taylor, 2000*) in its Matlab implementation, with a linear kernel. The specific method used to find the hyperplane was Sequential Minimal Optimization (SMO) (*Schölkopf and Smola, 2001*). The soft-margin parameter $C$ was set to its default value of 1 for all the trainings. Having found the hyperplane using the train data, the decoder was tested for its ability to correctly classify the remaining 1/5 of the response vectors (belonging to the test set), depending on the semi-space in which each vector was located.

This procedure was repeated for all possible combinations of the five subsets of response vectors in four train sets and one test set, with each combination yielding a distinct cross-validation loop. The outcomes of the classification, obtained across the resulting five cross-validation loops, were collected in a confusion matrix, which was then used to compute the performance of the decoder with that specific object pair in that specific subpopulation resampling iteration. The performance was computed both as the mutual information between the actual and the predicted object labels from the decoding outcomes and as the classification accuracy. As mentioned above, 50 resampling iterations were run for each object pair. The resulting 50 decoding performances were then averaged to

obtain the linear separability of the object pair. Finally, the performances obtained for different pairs were averaged to estimate the linear separability afforded by neuronal subpopulation.

As in the case of the single-cell analysis, only those pairs with objects having a luminosity ratio larger than a given threshold $Th_{LumRatio}$ (see Results) were included in the final average. Note that, in the case of the population analysis, this constraints had to be satisfied by all the neurons included in a given subpopulation, for all the areas. Obviously, the larger the subpopulation was (i.e., the larger $N$), the harder was to find pairs of objects that satisfied the constraint. For this reason, it was impossible to set $Th_{LumRatio} = 0.9$, as done in the single-cell analysis, because this choice would have yielded only a single object pair for the largest subpopulation (i.e., for $N = 96$). Therefore, for the analysis shown in the main text (*Figure 8C–F*), we set $Th_{LumRatio} = 0.8$, which yielded three object pairs for $N = 96$, and still allowed assessing linear separability in a regime where the luminance confound was kept under control. In addition, to check the generality of our conclusions, we also recomputed our analysis for $Th_{LumRatio} = 0.6$, which yielded 23 object pairs (*Figure 8—figure supplement 1A*).

## Test of generalization

The overall scheme of the decoding procedure was similar to the one described above (again, binary SVMs with linear kernels were used), with the key difference that, in any given subpopulation resampling iteration, only the response vectors of two randomly chosen transformations $t_x$ and $t_y$ of the two objects $o_i$ and $o_j$ were used to train the decoder. Following this training (i.e., once found the hyperplane), the decoder was tested for its ability to correctly classify (i.e., generalize to) the response vectors of all the remaining transformations (i.e., $T \neq t_x$ and $T \neq t_y$) of the same two objects, and the outcomes of this classification were collected in a confusion matrix. This procedure was repeated in such a way to choose exhaustively all possible combinations of train views $t_x$ and $t_y$. Since, in the population decoding analysis, 19 views per object were used (i.e., for each object, $T = \{t_1, t_2, \ldots, t_{19}\}$; see below for an explanation), this yielded $19^2$ confusion matrixes. The data from these matrixes were merged into a global confusion matrix, which was then used to compute the performance of the decoder with that specific object pair in that specific subpopulation resampling iteration. Again, the performance was computed both as the mutual information between the actual and the predicted object labels from the decoding outcomes and as the classification accuracy. The final generalization performance obtained for a given object pair was computed as the mean of the performances resulting from the 50 resampling iterations. Finally, the performances obtained for different pairs were averaged to estimate the generalization afforded by neuronal subpopulation.

As in the case of the linear separability analysis (see previous section), only those pairs with objects having a luminosity ratio larger than a given threshold $Th_{LumRatio}$ (see Results) were included in the final average. $Th_{LumRatio}$ was set to 0.8 (for the results shown in *Figure 8E–F*) and to 0.6 (for the results shown in *Figure 8—figure supplement 1B*), following the same rationale explained in the previous section.

## Computation of the chance level performance and of the linear separability of arbitrary groups of object views

As in the case of the single-cell analyses, we also computed chance classification performances (for both the tests of linear separability and generalization). These were obtained by running the same decoding analyses described above, but with the key difference of randomly shuffling first the association between object identity labels and pseudo-population response vectors. This was done following the same rationale of the single-cell analysis (see description above), i.e., by randomly shuffling only the label $O$, which specifies object identity, and not the label $T$, which specifies the transformation the objet underwent.

For any given neuronal subpopulation and any given object pair, chance performances were subtracted from the actual (i.e., un-shuffled) classification performances. This allowed measuring and reporting the net amount of transformation-tolerance afforded by the neuronal populations in the four visual areas. This subtraction was done only for the performances computed as the mutual information between the actual and the predicted object labels from the decoding outcomes (i.e., results shown in *Figure 8C and E*, left and *Figure 8—figure supplement 1A and B*, left). For the results reported in terms of classification accuracy (*Figure 8C and E*, right and *Figure 8—figure supplement 1A and B*, right), the chance performances were not subtracted from the actual performances,

so as to better highlight that the theoretical chance level was 50% correct discrimination. These chance performances were all between 49.4% and 50.1% correct discrimination for the linear separability test, and between 49.98% and 50.02% correct discrimination for the generalization test.

Finally, for each pair of objects $o_i$ and $o_j$, the linear separability analysis was also repeated for arbitrary groups of views of the two objects. To perform this test, in any subpopulation resampling iteration, out of the 19 views of object $o_i$, nine were randomly chosen and assigned to the first group, while the remaining 10 were assigned to the second group. Similarly, 10 randomly sampled views of object $o_j$ were assigned to the first group and the remaining nine to the second group. The decoding procedure was then performed as previously described. Only, in this case, the linear SVM decoders were trained to find a hyperplane in the response vector space that discriminated the set of views of the first group from the set of views of the second one. Fifty resampling iterations were carried out, each yielding a classification performance. The resulting 50 decoding performances were then averaged to obtain the linear separability of two arbitrary groups of object views, taken from that specific object pair. Finally, the performances obtained for different pairs were averaged to estimate the linear separability of arbitrary groups of object views afforded by neuronal subpopulation (shown as dashed lines in *Figure 8C* and *Figure 8—figure supplement 1A*).

## Choice of the object transformations used in the population decoding analyses

As explained in Materials and methods, out of the 38 transformations tested for each object, 23 were chosen, for each neuron, to be used in the single-cell decoding analyses. These included all the eight position changes, plus the remaining transformations (size and luminance changes, as well as in-plane and in-depth rotations) shown at the position that was closer to the neuron's RF center (i.e., either −15° or +15° of visual angle). Out of these 23 transformations, 19 were used in the population decoding analysis. In fact, for the neurons with RF center that was closer to −15°, the rightmost four positions (i.e., from +7.5° to +30° of visual angle) were excluded. Similarly, for the neurons with RF center that was closer to +15°, the leftmost four positions (i.e., from −22.5° to 0° of visual angle) were excluded. As a result, for the group of neurons with RFs closer to the −15° position, the 19 chosen transformations were the complementary set of the 19 transformations chosen for the group of neurons with RFs closer to the +15° position. This allowed aligning the two groups of neuronal RFs, by mapping the [−22.5°, 0°] range of positions of the first group onto the [ + 7.5°,+30°] range of positions of the second one, thus obtaining a single neuronal population, from which the neurons in each resampling iteration could be drawn.

## Acknowledgements

We thank Alessio Ansuini for his help in implementing the spike sorting procedure, Rosilari Bellacosa Marotti for her help with some of the statistical analyses and Alberto Petrocelli for his help in defining the protocols for anesthesia and surgery.

## Additional information

### Funding

| Funder | Grant reference number | Author |
| --- | --- | --- |
| Marie Curie International Reintegration Grant | PIRG6-GA-2009-256563 IVOR | Davide Zoccolan |
| Human Frontier Science Program | RGP0015/2013 | Davide Zoccolan |
| European Research Council | Consolidator Grant 616803-LEARN2SEE | Davide Zoccolan |
| ITN Marie Curie Grant | project ABC FP7-2007-2013/ PITN-GA-2011-290011 | Stefano Panzeri |
| Autonomous Province of Trento | Grandi Progetti 2012 "ATTEND" | Stefano Panzeri |

The funders had no role in study design, data collection and interpretation, or the decision to submit the work for publication.

## Author contributions
ST, Conceptualization, Resources, Data curation, Software, Formal analysis, Investigation, Visualization, Methodology, Writing—review and editing; HS, Conceptualization, Software, Formal analysis, Investigation, Visualization, Methodology, Writing—review and editing; GDF, Resources, Data curation, Formal analysis, Investigation, Visualization, Methodology; FBR, Resources, Data curation, Investigation; WV, Conceptualization, Software, Formal analysis, Visualization, Methodology, Writing—review and editing; MR, Resources, Formal analysis, Investigation, Visualization; FB, Resources, Formal analysis, Visualization; SP, Conceptualization, Formal analysis, Supervision, Funding acquisition, Investigation, Methodology, Writing—original draft; DZ, Conceptualization, Formal analysis, Supervision, Funding acquisition, Investigation, Visualization, Methodology, Writing—original draft, Project administration

## Author ORCIDs
Sina Tafazoli, http://orcid.org/0000-0003-1926-0227
Houman Safaai, http://orcid.org/0000-0002-8609-7397
Stefano Panzeri, http://orcid.org/0000-0003-1700-8909
Davide Zoccolan, http://orcid.org/0000-0001-7221-4188

## Ethics
Animal experimentation: All animal procedures were in agreement with international and institutional standards for the care and use of animals in research and were approved by the Italian Ministry of Health: project N. DGSAF 22791-A (submitted on Sep. 7, 2015) was approved on Dec. 10, 2015 (approval N. 1254/2015-PR); project N. 5388-III/14 (submitted on Aug. 23, 2012) and project N. 3612-III/12 (submitted on Sep. 15, 2009) were approved according to the legislative decree 116/92, article 7. All surgical procedures were performed under anesthesia, and every effort was made to minimize suffering (details explain din Materials and Methods).

# Additional files

### Major datasets
The following dataset was generated:

| Author(s) | Year | Dataset title | Dataset URL | Database, license, and accessibility information |
|---|---|---|---|---|
| Sina Tafazoli, Houman Safaai, Gioia De Franceschi, Federica Bianca Rosselli, Walter Vanzella, Margherita Riggi, Federica Buffolo, Stefano Panzeri, Davide Zoccolan | 2016 | Source data file for the article authored by Tafazoli and colleagues on invariant object representations in rat visual cortex | http://dx.doi.org/10.5061/dryad.vd8tf | Available at Dryad Digital Repository under a CC0 Public Domain Dedication |

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
