## [Decision Letter]

Thank you for submitting your article "Emergence of transformation-tolerant representations of visual objects in rat lateral extrastriate cortex" for consideration by *eLife*. Your article has been favorably evaluated by David Van Essen (Senior Editor) and three reviewers, one of whom is a member of our Board of Reviewing Editors. The following individual involved in review of your submission has agreed to reveal her identity: Nicole Rust (Reviewer #2).

The reviewers have discussed the reviews with one another and the Reviewing Editor has drafted this decision to help you prepare a revised submission.

Summary:

In this manuscript, the authors present a comparison of 4 visual areas in the rat with regard to object-based representations. The authors present a battery of stimuli consisting of various objects and identity-preserving transformations (size, position, view, etc.) of these objects and quantify using either mutual information or linear decoding analysis the extent to which single neurons and populations of neurons represent the visual stimulus in a transformation- invariant fashion. They find that while V1 is largely driven by low-level features (namely luminance in RF), neurons in higher visual areas LI and LL carry more information about high-level visual features (e.g. beyond total luminance). The work is systematic and carefully considered and the results can be compared with a better-understood species (the macaque). As such, the manuscript has the potential to become foundational for the field. The reviewers appreciate the care with which the authors have considered how simple low-level (luminance) modulations spuriously impact quantifying the responses of neurons at different stages of the pathway. Overall, the reviewers agree this is a high-quality study with interesting results, providing compelling evidence for object recognition function in lateral higher visual areas in rats.

The reviewers feel that the mutual information analysis should be included, but it needs to be better motivated. Furthermore, the authors need to take into account different total information about object identity (regardless of whether this is transformation-tolerant) across areas, in order to justify their claim that LL has a more linearly separable representation of object identity as a result of a transformation in information format from lower to higher stages.

Essential revisions:

1) The mutual information analysis is confusing to wade through. I appreciate that it represents upper bounds on the information present, however I think a more common method of making arguments about the representation of information in specific visual areas lies in decoding analysis or linear separability which is presented in later Figures (5 and 6). The problem with mutual information calculation is that I don't have a good metric with which to compare it to unlike decoding analysis where I know what chance decoding performance should be. The authors should explain the motivation for the mutual information analysis better, and perhaps compare to results of similar analyses in primates.

2) I have concerns related to the analyses that lead up to the main claim that area LL has a more robust view-invariant object representation than V1 (or LI). Figure 4 presents an analysis that deconstructs total information into 1) "view-invariant information", which is a quantification of how non-overlapping the responses to two objects are across views (regardless of whether that information is linearly separable) and 2) a component that accounts for the remaining non-overlap, which must be attributed to modulations by view. The main result that the authors highlight from Figure 4 is that for neurons that are selective for object features (not just luminance modulations), object information increases from V1 – LI – LL. Based on the claims outlined later in the paper, either the authors or I are misinterpreting what their "view-invariant information" measure reflects. As I understand it, 1) this is a "per neuron" (rather than total) measure of the amount of object-based information at each stage, and it is likely that different stages have different numbers of neurons, 2) it is a measure of the total information about object-identity, and 3) information about object-identity cannot be created in feed-forward pathway. The issue at hand is that – in the last paragraph of the subsection “The amount of view-invariant object information gradually increases from V1 to LL”, it implies that LL somehow has more total information for discriminating object identity than V1 (and LI) once luminance modulations are disregarded. This can only be true if 1) all the object-based information present in LL arrives there via a route that goes through V1 (and then LI) AND V1 (and LI) have more neurons than LL or 2) object-based information arrives in LL via some path that does not also pass through V1. Similarly, the linear separability analyses presented later in the paper (e.g. Figure 5 and Figure 8) are interesting only after first equating total information for object identity at each stage by (e.g.) allowing V1 (and LI) to have more neurons than LL. Once that is done properly, the main result – that LL is better at object discrimination with a linear read-out – may still hold (e.g. as a back-of-the-envelope calculation, LL has ~2x more object information per neuron (Figure 4) but comparing 96 V1 neurons to 48 LL neurons reveals that LL is still better (Figure 8). But this issue should be carefully considered for all analyses throughout the paper.

3) The major results were generated by computing the mutual information and training a decoder. These are good analytical tools, but do not offer intuitive interpretations for a broad audience (e.g., bit values << 1). To address this issue, it would be nice to provide some basic statistics about stimulus representation. From the example neuron in Figure 2, it seems the response of the LL neuron is sparser than the V1 neuron. Is this true at the population level? Since the information and decoding analysis indicate that LL is transformation tolerant, can the authors infer what neurons might be "encoding"/"tuned for"/"computing"? Did the authors see clues in firing rate variation over different transformations? A large transformation space was used, but we did not learn much about the representation of these transformations. Figure 4 imply that information about transformations decreased along the ventral stream. How was the transformation-tolerance generated along the ventral stream? Was the information about transformation along different axes reduced simultaneously or sequentially? Neuronal responses from four visual areas along the ventral pathway was collected. Other than V1 and LL, the author did not assign potential functions to LM and LI. Do they serve roles for transformation representations along particular dimensions?

4) One of the careful steps of the analysis was to control for luminance. The total information about the object stimuli declined along the ventral stream. It was when discriminating stimuli matched for RF and luminance, view-invariant information and linear discrimination accuracy increased along the ventral stream. It is understandable that at single neuron level the information about the stimuli is largely affected by the relative location of RFs and stimulus. But at the population level, it is not clear why it is necessary to limit the analysis to luminance-matched pairs. At the population level, each individual stimulus would be represented at various RF/luminance levels in different neurons. Thus, if the representation space was invariant to transformation, luminance control might not be needed. Can the authors obtain similar results at the population level without controlling so carefully for luminance?

5) The authors do a suitable job of controlling for overall luminance across the different object conditions, however neurons in V1 are likely not only responding to contrast defined by the object edge which is effectively what is being quantified here, but also the contrast on the object surface. It seems at first glance to be similar across the objects, but it would be nice to see a quantification of the contrast present on the object surfaces as this could also be driving firing rate differences across the object conditions.

6) To motivate luminance-matched analysis, the author showed that the information of luminance was reducing from V1 to LL. Is LL luminance insensitive to the same object? I.e., firing in LL should change less than in other areas to transformation along the luminance axis. This can be directly tested using the existing data set.

7) The calculation of RF luminance is affected by the shape of the stimulus and RF. Was the increase of f_high purely a result of larger RFs in LL? RFs in higher visual areas cannot be modeled simply by a single Gaussian profile. And RF shape can be affected by the stimuli (even in V1, e.g., Yeh, Xing, et al. 2009 PNAS). Is the calculation of RF luminance robust enough? Can the uncertainty affect the information and decoding analysis?

8) In Figure 4, we can see that V1, LM and LI actually have higher view-invariant object information when the luminance ratio is < ~0.5. Although it is arguable that the object pair may be discriminated by the difference in luminance, it is possible that luminance difference across transformation of the same object is larger than the luminance difference between the object pair. The later suggests transformation-tolerant object representation in these areas. Please explain/clarify.

9) The authors emphasize the similarities between the hierarchical organization of macaque and rat ventral temporal cortex, in that greater linear separability of different object classes, invariant to view, is seen in higher areas. However, I don't see any direct quantification of performance between the two species. The decoding performance in the rat seems significantly worse than that in monkey ITC (60% on a binary choice in the generalization task, where macaque ITC decoding performance reaches human levels). Please comment on how the values of discriminability and generalizability presented in Figure 7 compare to such analysis in the primate ventral stream.

---

## [Author Response]

*Essential revisions:*

*1) The mutual information analysis is confusing to wade through. I appreciate that it represents upper bounds on the information present, however I think a more common method of making arguments about the representation of information in specific visual areas lies in decoding analysis or linear separability which is presented in later Figures (5 and 6). The problem with mutual information calculation is that I don't have a good metric with which to compare it to unlike decoding analysis where I know what chance decoding performance should be. The authors should explain the motivation for the mutual information analysis better, and perhaps compare to results of similar analyses in primates.*

Following the reviewers’ suggestion, we have added two introductory paragraphs (subsection “The fraction of energy-independent stimulus information sharply increases from V1 to LL”, fourth paragraph and subsection “The amount of view-invariant object information gradually increases from V1 to LL”, first paragraph) to extensively explain the motivation of the mutual information analysis, its relationship with linear decoding analysis and the complementary nature of the two approaches. We have also consolidated the description of the information theoretic analysis, moving the definition of mutual information from the Methods to the Results (see eq. 1 and subsection “The fraction of energy-independent stimulus information sharply increases from V1 to LL”, fifth paragraph). Finally, we have also added a cartoon to Figure 3, to make it easier to interpret the meaning of mutual information (see Figure 3).

In general, while we appreciate that the use of information theoretic metrics is not very common in the study of visual object recognition, as compared to other approaches based on pattern classification, the information metrics are the most widely used in many other fields of sensory coding, following the influential work of Bialek, Richmond, Victor and many others. The use of information theory in this paper, besides being specifically suited to the particular nature of the problem we investigate, may help both this paper and the problem of object recognition to get more attention from a wider section of the computational neuroscience community, which, in turn, can be stimulated to bring more techniques and ideas into this research field.

*2) I have concerns related to the analyses that lead up to the main claim that area LL has a more robust view-invariant object representation than V1 (or LI). Figure 4 presents an analysis that deconstructs total information into 1) "view-invariant information", which is a quantification of how non-overlapping the responses to two objects are across views (regardless of whether that information is linearly separable) and 2) a component that accounts for the remaining non-overlap, which must be attributed to modulations by view. The main result that the authors highlight from Figure 4 is that for neurons that are selective for object features (not just luminance modulations), object information increases from V1 – LI – LL. Based on the claims outlined later in the paper, either the authors or I are misinterpreting what their "view-invariant information" measure reflects. As I understand it, 1) this is a "per neuron" (rather than total) measure of the amount of object-based information at each stage, and it is likely that different stages have different numbers of neurons, 2) it is a measure of the total information about object-identity, and 3) information about object-identity cannot be created in feed-forward pathway. The issue at hand is that in the last paragraph of the subsection “The amount of view-invariant object information gradually increases from V1 to LL”, it implies that LL somehow has more total information for discriminating object identity than V1 (and LI) once luminance modulations are disregarded. This can only be true if 1) all the object-based information present in LL arrives there via a route that goes through V1 (and then LI) AND V1 (and LI) have more neurons than LL or 2) object-based information arrives in LL via some path that does not also pass through V1.*

We thank the reviewers for pointing out the importance of being very clear about the fact that results such as those shown in Figure 4 should not be interpreted as implying that the total information about object identity present in each individual area increases along the visual hierarchy. We agree with the reviewers that, to address this point, the best way is to stress that the measure reported in Figure 4 is a *per neuron* measure, and not a *per area* measure. And, as correctly suggested by the reviewers, information per neuron can increase along the processing hierarchy, provided that the number of neurons decreases from one area to the next or that the hierarchy is not strictly feedforward. As it happens, this is exactly the case of rat visual cortical areas, whose size undergoes a dramatic and progressive shrinkage from V1 to LL, and whose connectivity is far from being purely feed-forward. In summary, the results of Figure 4 do not violate the data processing inequality. Rather, they indicate that rat lateral visual areas, while progressively losing neurons, concentrate more information in individual units, as a part of the process of making object information more explicit and more easily readable by downstream neurons that only have access to a limited number of presynaptic units. In our revised manuscript, we have added a paragraph to the Discussion to extensively comment on this issue (see subsection “Validity and implications of our findings”, sixth paragraph). In addition, we have also specified very clearly the “per-neuron” nature of our metrics in the text and in the axis labels of the figures, whenever appropriate (see our revised Figure 3–Figure 7).

*Similarly, the linear separability analyses presented later in the paper (e.g. Figure 5 and Figure 8) are interesting only after first equating total information for object identity at each stage by (e.g.) allowing V1 (and LI) to have more neurons than LL. Once that is done properly, the main result – that LL is better at object discrimination with a linear read-out – may still hold (e.g. as a back-of-the-envelope calculation, LL has ~2x more object information per neuron (Figure 4) but comparing 96 V1 neurons to 48 LL neurons reveals that LL is still better (Figure 8). But this issue should be carefully considered for all analyses throughout the paper.*

We agree that it is interesting to check if LL is still superior to the other areas (in terms of population decoding), after having approximately equated the populations under comparison, based on the amount of view-invariant information coded by the individual neurons in each population. To address this issue, we have carried out the analysis suggested by the reviewers, i.e., we have compared an LL population of 48 units to V1, LM and LI populations of 96 units, showing that LL is, indeed, still better than the other areas. We have added a new supplementary figure (Figure 8—figure supplement 3) to report this comparison and we have carefully explained the results of this new analysis and its motivation in the last paragraph of the subsection “Linear readout of population activity confirms the steep increase of transformation-tolerance along rat lateral visual areas”.

At the same time, we do not think that the population decoding results are interesting *only* after matching the total information across the areas. We believe that there is a value also in comparing populations of equal size, as done in most monkey ventral stream studies. In fact, the increase of invariant information per neuron along the areas’ progression is a property that any downstream neuron would necessarily exploit, while trying to read out object identity. Therefore, when comparing different areas in terms of their ability to support invariant recognition, it is important to take into account also differences among the invariant object information carried by individual neurons. This requires comparing the four areas at the population level by taking neuronal ensembles of the same size (as shown in Figure 8). We thus decided to present both the same-size comparisons already present in the previous version of the manuscript and the new information-matched analysis suggested by the reviewers.

*3) The major results were generated by computing the mutual information and training a decoder. These are good analytical tools, but do not offer intuitive interpretations for a broad audience (e.g., bit values << 1). To address this issue, it would be nice to provide some basic statistics about stimulus representation. From the example neuron in Figure 2, it seems the response of the LL neuron is sparser than the V1 neuron. Is this true at the population level?*

We agree with the reviewers that the information and decoding metrics may be somewhat hard to digest for a broader audience. On the one hand, we have tried to illustrate theseanalyses as clearly as possible, with the aid of a few single cell examples (Figure 2) and several cartoons (Figure 4, Figure 5, Figure 6 and Figure 8). We have also added a new cartoon to Figure 3 (see the new Figure 3) to better explain the meaning of the mutual information analysis. We believe that these cartoons substantially help making our analyses and results intelligible. On the other hand, given that our manuscript is focused on neuronal coding of high-level object information, resorting to simpler (more intuitive) metrics or analyses would not be appropriate – it would not allow capturing the essence of the information rat visual areas encode, and could potentially lead to misleading conclusions.

A nice example of the limitation of more intuitive metrics is exactly the sparseness measure that the reviewers mentioned. It is true that, for the example neurons shown in Figure 2, the LL unit fires more sparsely than the V1 unit. However, at the population level, LL neurons did not fire more sparsely than V1 neurons, but this result should not be taken as an indication of V1 and LL processing the visual input in a similar way. In fact, sparseness is a metric that is affected by both object selectivity (which tends to make sparseness higher) and transformation tolerance (which tends to make sparseness lower). As shown by Rust and DiCarlo (2012), although these properties both increase across the monkey ventral stream (i.e., from V4 to IT), sparseness remains unchanged. Therefore, sparseness cannot be taken as a very informative measure to compare visual object representations along a putative ventral- like pathway. We have added a paragraph to our revised manuscript, where we explain this and we report the sparseness comparison between V1 and LL at the population level (see subsection “The fraction of energy-independent stimulus information sharply increases from V1 to LL”, third paragraph).

*Since the information and decoding analysis indicate that LL is transformation tolerant, can the authors infer what neurons might be "encoding"/"tuned for"/"computing"?*

We agree with the reviewers that understanding what combination of visual features any given neuron is tuned to is a very interesting topic. In primate studies, this endeavor has a very long tradition, starting from the reduction method of Tanaka and colleagues till the more refined and quantitative efforts of Connor’s group. All these studies rely on a very specific design, selection and parameterization of the stimuli (Connor), and make use of model fitting procedures (Connor) that can only be applied to data recorded using large, parametric shape spaces. Our stimulus set was not designed for applying such approaches and, as a result, a parametric analysis of what visual features each of our neurons was tuned to is not possible. The reason is that, by design, our work belongs to a different family of studies, whose goal is to measure the ability of single neurons and neuronal populations to code invariantly object identity, and not to infer shape tuning itself. Nevertheless, our study does provide some very careful quantification of the extent to which two low-level visual properties are coded across rat lateral visual areas: stimulus luminance and, in our revised manuscript, also stimulus contrast (which was computed to address the reviewers’ essential revision #5; see the new Figure 3—figure supplement 4).

*Did the authors see clues in firing rate variation over different transformations? A large transformation space was used, but we did not learn much about the representation of these transformations. Figure 4 imply that information about transformations decreased along the ventral stream. How was the transformation-tolerance generated along the ventral stream? Was the information about transformation along different axes reduced simultaneously or sequentially?*

In our manuscript, we have carried out most of our analyses by including all the transformations at once (along all variation axes). The motivation was to probe invariance under the most challenging conditions, to allow the differences among the visual areas to emerge more clearly. However, we agree with the reviewers that presenting the transformation-tolerance attained by the various areas with regard to specific axes is also important. This is why, in Figure 6 we have presented the results of the most important decoding analysis (where generalization across transformation was tested) across all four variation axis: position, size, luminosity and rotation. We believe that this figure already answers many of the questions of the reviewers. In fact, it can be seen how LL is always the area attaining the largest transformation tolerance, typically followed by LI and then by LM and V1. In other words, the same growth of invariance observed when all transformations are considered together (Figure 6), is visible also for the individual variation axes (Figure 6). This means that tolerance was built up gradually along the V1, LI, LL pathway and “simultaneously” along all tested transformation axes. We apologize if this analysis – already present in the first submission – might have gone unnoticed by the reviewers, as it was described very briefly. In our current revision, we have expanded this section and better emphasized this result (see subsection “Object representations become more linearly separable from V1 to LL, and better capable of supporting generalization to novel object views”, fourth paragraph).

In addition, we have carried out two new analyses, in which we have measured position tolerance and size tolerance in the four areas across parametric changes. These new analyses confirmed the superior tolerance achieved by LL, thus fully addressing, we believe, the reviewers’ concern (these new analyses are shown in the new Figure 6—figure supplement 1 and are described in the fifth paragraph of the aforementioned subsection, with a detailed explanation provided in the Methods subsection “Test of generalization along parametric position and size changes”).

*Neuronal responses from four visual areas along the ventral pathway was collected. Other than V1 and LL, the author did not assign potential functions to LM and LI. Do they serve roles for transformation representations along particular dimensions?*

In all our analyses, we have always compared all four areas. In our discussion of the results, we often focused on LL, and its difference from V1, partly because these two areas are at the extremes of the anatomical progression, and partly (and more importantly) because our data show that they also behave as opposite poles of the functional hierarchy. However, we have not overlooked the implications of our findings with regard to LM and LI. Most of our analyses show that LI behaves as an intermediate processing stage in the functional hierarchy (for instance, this is clearly stated in the conclusions we draw at the beginning of the Discussion (third paragraph), yielding view-invariant information and decoding performances that are significantly larger than V1 and LM (Figure 4–Figure 8), although often significantly lower than LL (Figure 5 and Figure 8). So, we did assign a specific function to LI as an intermediate object processing stage, between V1 and LL. With regard to LM, we have commented extensively in the Discussion that its functional properties (in terms of invariant coding) do make it indistinguishable from V1. We have also made reference to the anatomical and functional literature of the mouse visual system, showing that this area has mixed ventral-like and dorsal-like properties, thus suggesting that LM can be homologous to area V2 in primates (which belongs to both pathways). We believe that our data support this interpretation (this is clearly stated in the third paragraph of the Discussion).

*4) One of the careful steps of the analysis was to control for luminance. The total information about the object stimuli declined along the ventral stream. It was when discriminating stimuli matched for RF and luminance, view-invariant information and linear discrimination accuracy increased along the ventral stream. It is understandable that at single neuron level the information about the stimuli is largely affected by the relative location of RFs and stimulus. But at the population level, it is not clear why it is necessary to limit the analysis to luminance-matched pairs. At the population level, each individual stimulus would be represented at various RF/luminance levels in different neurons. Thus, if the representation space was invariant to transformation, luminance control might not be needed. Can the authors obtain similar results at the population level without controlling so carefully for luminance?*

We thank the reviewers for this comment, which has helped us clarifying why applying the constraint over the luminance of the objects to be decoded is essential also at the population level.The fact is that, in general, a neuronal population will inherit its discrimination power from its constituent neurons. And this applies also to the coding of luminance. If luminance differences can be used as a cue, by single neurons, to solve the invariant task, because of the sharp tuning of single neurons for luminance, then the neuronal population will retain this ability. To show that this was the case, we have run the population analyses of Figure 8 for the full spectrum of Th_LumRatio. The resulting performance curves show how the performance itself is extremely sensitive to the difference of luminance that the objects are allowed to have, displaying a trend that is similar to what already reported for single neurons in Figure 4. Hence, the need to control for the luminance confound also in the population analysis. We have now better explained this in the fifth paragraph of the subsection “Linear readout of population activity confirms the steep increase of transformation-tolerance along rat lateral visual areas” and we have reported the result of this new analysis in the new Figure 8—figure supplement 2.

*5) The authors do a suitable job of controlling for overall luminance across the different object conditions, however neurons in V1 are likely not only responding to contrast defined by the object edge which is effectively what is being quantified here, but also the contrast on the object surface. It seems at first glance to be similar across the objects, but it would be nice to see a quantification of the contrast present on the object surfaces as this could also be driving firing rate differences across the object conditions.*

We thank the reviewers for this very useful suggestion. We have carried the analysis they suggested. To this aim, we have defined a new metric, which we called RF contrast, that quantifies the variation of luminance produced by any given stimulus within the RF of a neuron. We have then applied the same analysis based on mutual information (i.e., same analysis of eq. 2) to quantify how much information neurons in the four areas carried about RF contrast and the complementary information that was not about RF contrast. The results of this new analysis are shown in a new supplementary figure (Figure 3—figure supplement 4) and are described in the last paragraph of the subsection “The fraction of energy-independent stimulus information sharply increases from V1 to LL” (see also the Methods subsection “Computation of the Receptive Field (RF) luminance and RF contrast”, last paragraph). We have found, for stimulus contrast, a trend that was very similar to the one found for stimulus luminance, with the information about RF contrast decreasing monotonically along the areas progression. As a consequence, the fraction of contrast-independent information increased monotonically and significantly from V1 to LL. Overall, this new analysis confirms the pruning of low-level information that takes place along rat lateral visual areas.

*6) To motivate luminance-matched analysis, the author showed that the information of luminance was reducing from V1 to LL. Is LL luminance insensitive to the same object? I.e., firing in LL should change less than in other areas to transformation along the luminance axis. This can be directly tested using the existing data set.*

The intuition of the reviewers is correct. To show this, we have performed a new analysis, in which we report the tuning to luminance variation of the same object for the neurons in the four visual areas. This tuning is sharper in the more medial areas (V1 and LM) than in the more lateral ones (LI and LL). We have quantified this trend by computing the sparseness of the luminance tuning curves, and we have found that sparseness decreases progressively and monotonically from V1 to LL. The results of this new analysis are shown in a new supplementary figure (Figure 3—figure supplement 3) and are described in the eighth paragraph of the subsection “The fraction of energy-independent stimulus information sharply increases from V1 128 to LL”.

*7) The calculation of RF luminance is affected by the shape of the stimulus and RF. Was the increase of f_high purely a result of larger RFs in LL?*

We have controlled for this, by running the analysis shown in Figure 7, where we have taken subpopulations in all 4 areas with matched RF sizes. Figure 7 shows that the growth of f_high is as strong as for the full populations. So, the increase of f_high does not depend on RF size.

*RFs in higher visual areas cannot be modeled simply by a single Gaussian profile. And RF shape can be affected by the stimuli (even in V1, e.g., Yeh, Xing, et al. 2009 PNAS). Is the calculation of RF luminance robust enough? Can the uncertainty affect the information and decoding analysis?*

This is a very interesting topic, which definitely deserves some in-depth discussion. We have now added a full paragraph to deal with this issue (subsection “Validity and implications of our findings”, second paragraph).

Briefly, with regard to the dependence of the RFs on the type of stimuli used to map them, in our experiments, we used very simple stimuli (high contrast drifting bars), with the goal of measuring the luminance-driven component of the neuronal response, so as to estimate what portion of the visual field each neuron was sensitive to. This is a very simple but effective procedure, because all the recorded neurons retained some amount of sensitivity to luminance, even in LL (see Figure 3). As a result, it was possible to obtain very clean RF maps in all the areas (as shown by the examples in Figure 1).

With regard to the accuracy of fitting RF maps with 2-d Gaussians, we agree that, in general, such fits may not be appropriate for all visual neurons, but we do not believe that this procedure cannot be applied in higher-level areas. In fact, fitting 2-d Gaussians is commonly accepted as a valid procedure to get a first approximation of the RF extension/shape, even in high-level visual areas, such as monkey IT (e.g., see Op De Beeck and Vogels, 2000; Brincat and Connor, 2004; Rust and DiCarlo, 2010). In any event, in our study, we took two measures to avoid possible mistakes/biases, produced by inaccurate Gaussian fits. In the RF size analysis (Figure 7), we only included data from neurons whose RFs were very well fitted by Gaussians. More importantly, for the computation of the RF luminance, we did not use the fitted RFs, but we directly used the raw RF maps. This allowed weighting the luminance of the stimulus images using the real shapes of the RFs, and allowed computing the RF luminance for all the neurons, also those with RFs that were not well fitted by Gaussians.

*8) In Figure 4, we can see that V1, LM and LI actually have higher view-invariant object information when the luminance ratio is < ~0.5. Although it is arguable that the object pair may be discriminated by the difference in luminance, it is possible that luminance difference across transformation of the same object is larger than the luminance difference between the object pair. The later suggests transformation-tolerant object representation in these areas. Please explain/clarify.*

To address the concern of the reviewer, we have carried out a new analysis, whose result is reported in a new supplementary figure (Figure 4—figure supplement 1). In this analysis, we have computed the information that RF luminance is able to convey about object identity in spite of transformations: I(L;O). This information is very large in all the areas, unless the objects in the pairs are forced to have similar luminance (by requiring the luminance ratio to approach 1), in which case I(L;O) drops sharply to zero. The crucial fact is that, for V1, LM and LI, the drop of I(L;O) as a function of the luminance ratio follows the same exact trend as the drop of the invariant object information I(R;O) shown in Figure 4. This implies that, in these areas, a large fraction of the invariant information is indeed accounted for by the luminance differences of the object pairs. In turn, this proves the need of restricting the computation of I(R;O) to the largest possible value of the luminance ratio. Our new analysis shows that setting the constraint on the luminance ratio to 0.9 effectively get rid of the luminance confound in all the areas, thus guaranteeing a fair comparison. This new analysis is described in the sixth paragraph of the subsection “The amount of view-invariant object information gradually increases from V1 to LL”.

*9) The authors emphasize the similarities between the hierarchical organization of macaque and rat ventral temporal cortex, in that greater linear separability of different object classes, invariant to view, is seen in higher areas. However, I don't see any direct quantification of performance between the two species. The decoding performance in the rat seems significantly worse than that in monkey ITC (60% on a binary choice in the generalization task, where macaque ITC decoding performance reaches human levels). Please comment on how the values of discriminability and generalizability presented in Figure 7 compare to such analysis in the primate ventral stream.*

We thank the reviewer for this suggestion. Although a direct quantitative comparison of decoding performances between monkeys and rats would be of great interest, we feel that a quantitative comparison between published experiments in primates and our new experiment in rodents would be difficult to interpret, because of the large differences in presentation protocols and combinations of object identity and transformations across these studies. In fact, even comparing object representations in different areas of the same species (the macaque) has required recording those areas (e.g., V4 and IT) under the same exact experimental conditions (e.g., see Rust and DiCarlo, 2010). In the case of our study, the reviewers point to the fact that a 60% generalization performance is not very high, but this performance was achieved in a test of generalization from a single object view to 19 other different views, spread across four different translation axes, while most monkey IT studies have tested generalization to single steps along individual transformation axes.

However, thanks to the reviewers’ comment, we have realized that, in our previous version of the manuscript, it was not sufficiently clear that one of our primary goals was achieving a qualitative comparison among rats and monkeys in terms of the existence of a progression of visual cortical representations to support invariant recognition. These qualitative comparisons are still possible, and lend themselves to useful interpretations, even in the face of the differences in the experimental protocols applied so far across species. We have now amended the text, adding a new section to the Discussion to clearly explain this (second paragraph).